# Cancer genomes tolerate deleterious coding mutations through somatic copy number amplifications of wild-type regions

Fabio Alfieri [1], Giulio Caravagna[2] & Martin H. Schaefer [1] ✉

Cancers evolve under the accumulation of thousands of somatic mutations and chromosomal aberrations. While most coding mutations are deleterious, almost all protein-coding genes lack detectable signals of negative selection. This raises the question of how tumors tolerate such large amounts of deleterious mutations. Using 8,690 tumor samples from The Cancer Genome Atlas, we demonstrate that copy number amplifications frequently cover haploinsufficient genes in mutation-prone regions. This could increase tolerance towards the deleterious impact of mutations by creating safe copies of wild-type regions and, hence, protecting the genes therein. Our findings demonstrate that these potential buffering events are highly influenced by gene functions, essentiality, and mutation impact and that they occur early during tumor evolution. We show how cancer type-specific mutation landscapes drive copy number alteration patterns across cancer types. Ultimately, our work paves the way for the detection of novel cancer vulnerabilities by revealing genes that fall within amplifications likely selected during evolution to mitigate the effect of mutations.

Cancer genomes evolve through the acquisition of multiple types of somatic aberrations – these include somatic single nucleotide variants (SNVs or mutations) and copy number alterations (CNAs). A small fraction of these thousands of aberrations[1,2] is considered beneficial from the tumor perspective (driver events), while the vast majority is not undergoing detectable selection[3–6] (passenger events). However, in many systems – including cancer cells – the majority of coding mutations have been observed to have a deleterious role on cellular fitness: e.g., in silico evolutionary simulations[7], in vitro and in vivo proliferation assays[8–10], and whole-genome CRISPR-KO screens within different species and many tumor cell lines[11], showed a reduced in vitro proliferative fitness and a slower in vivo tumor growth, demonstrating the deleterious role of most coding passenger mutations. Specifically, among point mutations in coding regions, more than 75% are non-synonymous[12], and those mildly or strongly deleterious for cellular fitness are predicted to be within a range of about 65%[13] to >70%[14].

While the extent of negative selection acting on cancer genomes is still under debate[3–6,15–19], common measures of selection – such as the ratio of non-synonymous to synonymous substitution rates (dN/dS) – indicate that most protein-coding genes lack detectable signals of negative selection in cancer genomes: as a matter of fact, 99% of coding mutations appear to be tolerated[5,16]. This contrasts with negative selection during germline (organismal) evolution, where the majority of genes are under negative selection and thereby protected from mutations[20–22]. These observations raise the question of how cancers can tolerate many deleterious mutations.

Typically, two complementary answers are given to this question. First, most mutations might have deleterious effects only on the development of an organism (germline evolution) but are instead tolerated in differentiated tissues or tumors (somatic evolution). This is surprising given that many of the functions under negative selection in germline evolution affect general processes that should be important in every cell of the body, in particular in fast-dividing cancer cells.

[1]Department of Experimental Oncology, IEO European Institute of Oncology IRCCS, Milan 20139, Italy. [2]Department of Mathematics and Geosciences, University of Trieste, Trieste 34127, Italy. ✉e-mail: martin.schaefer@ieo.it

E.g., globally expressed genes are under negative selection in the germline[20,23]. This indicates that general housekeeping functions (which should be important in somatic lineages too) are protected from mutations. Also, proliferation assays performed in cancer cells demonstrated that losing most genes would have a detrimental fitness effect[11].

The second explanation for the lack of detectable negative selection – though not independent from the first one – could be a consequence of the asexual nature of cancer evolution: in the absence of recombination, populations irreversibly accumulate mutations, and cells rapidly experience a fitness decrease - a process called Muller's ratchet[24]. In populations of finite size, with large genomes and high mutation rates, the ratchet-like accumulation of deleterious mutations increases the likelihood of population extinction through mutational meltdown[25–27]. This again suggests that tumors may evolve strategies to buffer for deleterious mutations. Diploidy, which is the presence of two copies of each gene, is certainly the most important evolutionary invention providing protection against mutation-induced gene inactivation[5,28]. However, in certain cases where the overall mutation load is higher – such as in cancer cells – diploidy might not be sufficient to buffer deleterious mutations[29].

Nonetheless, we here propose a third complementary answer: cancer somatic copy-number amplifications can reinforce the tolerance towards deleterious mutations, countering the phenomenon of Muller's ratchet. During cancer evolution, CNAs are randomly acquired in the genome, and when they temporally occur before somatic mutations, they create intact copies of the genes therein. Those amplifications can protect against the subsequent deleterious and irreversible accumulation of mutations, and then they would be selected and over-represented in mutation-prone regions with cancer-relevant genes. Similar phenomena have already been observed in different species such as the asexual amoebas, yeast, and bacteria: the firsts gain additional chromosomes through polyploidy, which favors gene conversion of mutated copy using wild-type ones[30,31], while the latter use horizontal gene transfer to restore inactivated genes[32]. Another example of relaxed selection through homology is paralogs (two or more DNA segments having shared ancestry), which provide redundancy and compensate for gene loss or inactivation. In fact, genes without paralogs or with only one paralog undergo more effective negative selection than genes with multiple paralogs[4]. This is in line with phylogenetic studies suggesting that gene duplication leads to relaxed purifying selection during species evolution[33]. Notably, in cancer cells, whole-genome doubling – which preserves the ratio of mutant to wild-type alleles while doubling the number of wild-type ones – has been recently suggested to serve as a compensatory phenomenon for the accumulation of deleterious mutations in haploid loss-of-heterozygosity (LOH) regions[34].

Given all the previous observations, we sought to investigate the hypothesis that somatic amplifications are positively selected in cancer evolution as they can buffer the subsequent gene inactivation through loss-of-function (LOF) mutations in cancer-relevant regions of the genome. We investigated this buffering phenomenon through the analysis of the relationship between a copy-neutral mutation score – considering only mutations within copy-neutral regions – from now on the $\mu$ score, and the amplification frequency at multiple scales (genes, segments from 1 Mbp to 50 Mbp, chromosome arms and whole chromosomes). We specifically designed the $\mu$ score in order to avoid the trivial confounder between copy number and mutations: the more copies of a segment, the more mutations the segment can acquire. Therefore, after dividing the genome into segments with a specific length ($j$), only copy-neutral segments were considered for computing the number of mutations, thus normalized for the total number of coding bases and patients, as shown in Eq. (3) in "Methods" section and in Fig. S1.

We observe that intermediate-sized segment lengths provide best correlations between mutations and amplification frequency, suggesting that amplifications of around 36 Mbp could act as optimal buffers (by maintaining a balance between deleterious mutation accumulation and tumor fitness). We examine the properties of genes – such as haploinsufficiency, expression, or being a cancer gene – and the estimates of mutation functional impacts on the likelihood of genomic regions being buffered by amplifications. We demonstrate that regions highly susceptible to deleterious mutations and containing cancer-essential genes are often amplified during tumor evolution. This indicates that those amplifications might give a selective advantage to tumor cells to protect their genome against the detrimental effects of mutations and to maintain their fitness. By exploiting the buffering effect, we identify groups of genes that are protected, reflecting essential cellular properties. Accordingly, our study unveils a central factor that shapes cancer genomes during tumor evolution where different sources of molecular aberrations constrain and enable each other. Our work contributes to the understanding of how tumors can tolerate thousands of deleterious mutations.

## Results
### Genomic amplification events act as a buffer across genomic scales
We hypothesized that somatic copy number (CN) amplifications might buffer the deleterious effect of passenger mutations during tumor evolution (Fig. 1a). If this was the case, amplifications should occur more often in mutation-rich tumors and, specifically, in high mutation load regions. To test this, we performed a correlation analysis associating $\mu$ score and amplification burden within 23 TCGA cancer types from primary solid tumors (8,690 patients). The $\mu$ score was computed by considering mutations within copy-neutral regions only for the respective segment, therefore avoiding the obvious confounder that the more copies of a segment exist the more mutations the segment could acquire (see "Methods" section and Fig. S1). As expected, we observed a significant relationship between the amplification frequency and the $\mu$ score over cancer types (Fig. 1b).

If the amplifications acted as buffers, we would expect them to occur in mutation-rich regions. Therefore, we performed a segment-specific association analysis between the amplification frequency and the $\mu$ score. Across all 23 tumor types, 9 of them showed statistically significant positive correlations supporting our hypothesis: Lung Squamous cell Carcinoma (LUAD), Breast invasive Carcinoma (BRCA), Lung Squamous cell Carcinoma (LUAD), Glioblastoma multiforme and Brain Lower Grade Glioma (GBMLGG), Thyroid carcinoma (THCA), Colon and Rectum Adenocarcinoma (COADREAD), Pancreatic adenocarcinoma (PAAD), Cervical Squamous cell Carcinoma and Endocervical Carcinoma (CESC), and Head and Neck Squamous Carcinoma (HNSC); Supplementary Dataset 1 and Fig. 1c. We focused on those nine tumor types in the following analyses.

An alternative explanation for this association could be that different types of aberrations occur in mutation-prone regions: these segments can accumulate, in addition to SNVs, many double strand-breaks, which eventually lead to CNAs. This would potentially explain the observed association between amplifications and mutations. Thus, we tested whether we observed the same behavior between the $\mu$ score and the deletion frequency (deletions may also originate from double strand-breaks). We detected a positive but not significant trend between the $\mu$ score and the deletion burden at the tissue-type level (Fig. S2a). However, the segment-specific correlations exhibit a complete absence of association between the $\mu$ score and the deletion frequency (Fig. 1d). These results discarded that possible confounding explanation of an association between double strand breaks and SNVs; furthermore, they showed that, despite the overall CNA load correlates

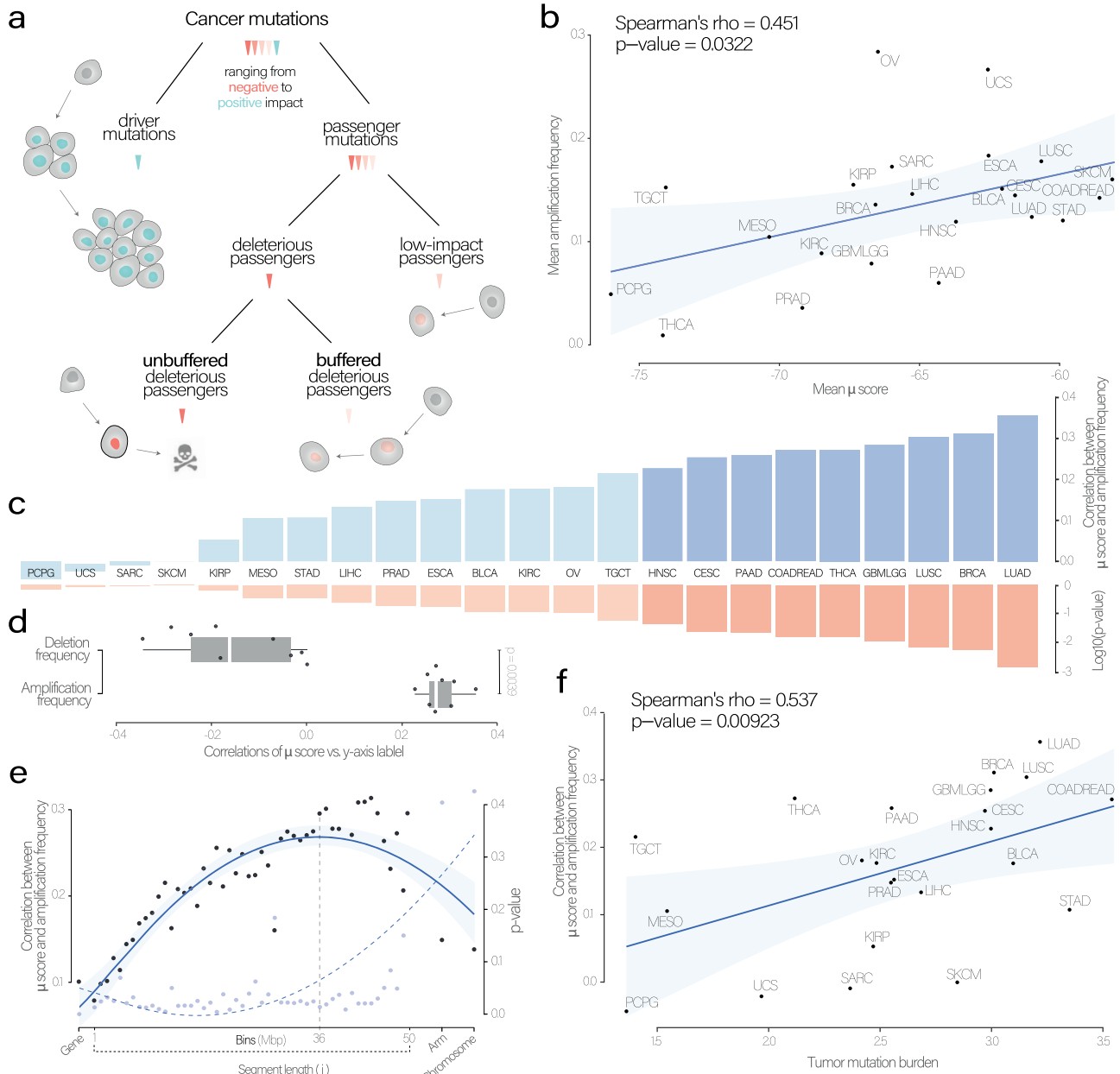

**Fig. 1 | Mutation-rich regions and genomes are often amplified in cancer.**
**a** Mutation classification according to the proposed evolutionary mechanism. In protein-coding regions different mutation types can be accumulated: very few of them can be classified as drivers (which have a strong positive impact on tumor proliferation), while the vast majority are passengers, the latter are considered to have neutral/mildly low or negative impact. Our hypothesis is that the negative fitness impact of deleterious passengers can be buffered if the genomic region is protected by amplification events (buffered deleterious passenger). **b** Spearman's correlation between the mean mutation ($\mu$) score and mean amplification frequency ($p$-value calculated using Spearman's correlation test). **c** Spearman's correlations between amplification frequency and the $\mu$ score in 23 analyzed cancer types with associated $p$-values (below), darker colors represent significant tumor types with Spearman's $p$-value $\leq 0.05$ (no multiple testing correction).
**d** Spearman's correlations between amplification frequency and $\mu$ score and between deletion frequency and $\mu$ score in $n = 9$ significant cancer types. The $p$-value was determined using a paired two-sided Wilcoxon test. The box represents the 1st to 3rd quartile with the median marked by a horizontal line. **e** From right to left Spearman's correlations were computed using different increasing segmentation sizes: from single genes, bins (range 1–50 Mbp), chromosome-arms up to entire chromosomes. Blue curve represents Spearman's rho, vertical dotted line

represents the chosen segment length ($j$) (36 Mbp), while dotted curve line represents the $p$-value trend. **f** Spearman's correlation (and test) between Spearman's correlation (between amplification frequency $\mu$ score) and mutation burden. For **b**, **e** and **f**, the error bands reflect the 95% confidence intervals of these estimates. Lung Squamous cell Carcinoma (LUSC); Lung Adenocarcinoma (LUAD); Colon Adenocarcinoma and Rectum Adenocarcinoma (COADREAD); Cervical Squamous cell Carcinoma and Endocervical Carcinoma (CESC); Breast invasive Carcinoma (BRCA); Skin Cutaneous Melanoma (SKCM); Ovarian Serous Cystadenocarcinoma (OV); Uterine Carcinosarcoma (UCS); Liver Hepatocellular Carcinoma (LIHC); Head and Neck Squamous Carcinoma (HNSC); Prostate Adenocarcinoma (PRAD); Thyroid carcinoma (THCA); Pheochromocytoma and Paraganglioma (PCPG); Esophageal carcinoma (ESCA); Stomach adenocarcinoma (STAD); Glioblastoma multiforme, Brain Lower Grade Glioma (GBMLGG); Kidney renal clear cell carcinoma (KIRC); Kidney renal papillary cell carcinoma (KIRC); Pancreatic adenocarcinoma (PAAD); Testicular Germ Cell Tumors (TGCT); Mesothelioma, (MESO); Sarcoma (SARC); Bladder Urothelial Carcinoma (BLCA). Source data are provided as a Source Data file. Cell representations were adapted from "Icon Pack - Cell Biology" (https://app.biorender.com/biorender-templates/) and the skull from the icon selection of BioRender.com (2023).

with the $\mu$ score, the amplifications often occur protecting highly mutated regions, while the deletions do not.

As the amplifications are associated with a fitness cost too[35–39], we wondered which is the segmentation length where we have an optimal trade-off between amplification fitness cost and its buffering-associated fitness gain. We correlated the $\mu$ score and the amplification frequency over varying segment lengths ($j$) ranging from gene level to chromosomes (Fig. 1e). $j = 36$ Mbp was the optimal segment length for further analyses since it displayed the best associations in terms of Spearman's correlation and associated $p$-value.

As we expected that the detrimental fitness cost of mutations would sum up, we tested if mutation-rich tumor types showed a stronger buffering effect. Indeed, we observed that tumor types with higher mutation burden are associated with the strength of our buffering estimates (the correlations between the $\mu$ score and the amplification frequency; Fig. 1f). In contrast, having a higher amplification burden does not necessarily indicate a strong correlation, as depicted in Fig. S2b. This suggests that amplifications are likely to have been selected during tumor evolution in patients with high mutation rates rather than mutation rates being elevated as a consequence of more genomic copies.

## Mutation type, gene function, and expression modulate the buffering effect

Mutation-induced LOF produces proteins with diminished, lost or altered functions with potentially dramatic effects on the fitness of a cell (Fig. 1a). We wondered if any specific properties of the mutated gene or the mutation type itself may affect our buffering estimates (the correlations between the $\mu$ score and the amplification frequency). We conducted all subsequent analyses on the nine cancer types that exhibited a statistically significant positive correlation (Fig. 1c).

We hypothesized that haploinsufficient genes (i.e., those genes where just one unmutated copy of the gene is insufficient to maintain its wild-type function) should be strongly prone to buffering events. Our relationship analysis demonstrated that frequently mutated haploinsufficient genes tend to be amplified during tumor evolution, which is likely to create a buffer to compensate for the loss of the gene product upon mutation. In all tested cancer types, we observed that correlations between the $\mu$ score and the amplification frequency increased when considering only haploinsufficient genes and decreased when considering only non-haploinsufficient genes (Figs. 2a and S3 for the cancer types with non-significant associations between $\mu$ score and amplification frequency). The differences in correlations between haploinsufficient and non-haploinsufficient genes were significant when testing for differences in correlation distributions over all cancer types using two scores as haploinsufficiency predictors: the probability of being loss-of-function intolerant (pLI[40]) score and the genome-wide haploinsufficiency score (GHIS[41]) (Fig. 2b; $p = 0.0039$ and $p = 0.0078$, respectively; paired two-sided Wilcoxon test). This finding highlights again the protective function of amplifications that mainly compensate for the deleterious mutations in genes that do not tolerate LOF mutation.

Next, we analyzed whether a notable discrepancy exists in the buffering of expressed versus non-expressed genes and whether this is reflected in the association between amplification frequency and the $\mu$ score. By integrating transcriptomic data into our analysis, we could discriminate which genes are actively transcribed. Our findings revealed a significant decrease in correlations for non-expressed genes in all analyzed cancer types, with the only exception of LUSC, as illustrated in Fig. 2c. Moreover, we observed a significant difference in the distribution of correlations (Fig. 2d; $p = 0.012$; paired two-sided Wilcoxon test) when comparing expressed and non-expressed genes.

After that, we focused on the impact of mutations themselves. We speculated that the sites predicted to have a low impact on gene products upon mutations should be less prone to be compensated by

amplifications. Specifically, we exploited mutation impact scores (Polyphen[42] and CADD[43]), which predict consequences on the protein in terms of structure and function, to classify either as lowly damaging or highly damaging. We observed a correlation drop when removing the highly damaging ones (Fig. 2e): this means that deleterious mutations are not well tolerated and, consequently, highly buffered. Our analysis using CADD as a mutation impact predictor revealed significant differences in correlation distributions among all considered cancer types (Fig. 2f; $p = 0.039$; paired two-sided Wilcoxon test). In contrast, when using Polyphen, we still observed a difference in correlation distributions, although the effect was less pronounced.

Similar to the previous analysis, we also investigated the effect of synonymous and non-synonymous mutations on the correlations between the amplification frequency and the $\mu$ score. Our analysis revealed a correlation drop for synonymous mutations in most of the analyzed cancer types (Fig. 2g), indicating that they are not as strongly buffered as non-synonymous mutations. This finding is consistent with the notion that synonymous mutations should be in general less impactful compared to non-synonymous mutations, as they do not alter the amino acid sequence of the encoded protein. In addition, we found a significant difference in the distribution of correlations when comparing synonymous and non-synonymous mutation classes (Fig. 2h; $p = 0.011$; paired two-sided Wilcoxon test).

Amino-acid changes or premature stop codons as result of non-synonymous mutations may cause a higher propensity of proteins to aggregate, usually exposing hydrophobic domains that are normally buried within their interiors. They are recognized by specific mitochondrial, cytosolic and endoplasmatic (ER)-sensors, mainly HSP70 family members, which trigger the cellular proteotoxic stress response. This cytoprotective response leads to cell cycle arrest, to a general inhibition of protein synthesis, and to potential proteotoxicity-associated cell death[44]. We reasoned that creating a further copy of an aggregation-prone region may increase the probability of mutating this region and hence induce cellular stress. Therefore, we computed the aggregation probability of mutated and wild-type sequences using TANGO[45] (Supplementary Dataset 2), speculating that amplifications should tend to mainly buffer LOF mutations but avoid aggregation-prone regions. Aggregation-causing mutations exhibited a substantial loss of buffering compared to non-aggregation-causing ones in all cancer types (Fig. 2i and Fig. 2j; $p = 0.0078$; paired two-sided Wilcoxon test).

As oncogene (OG) and tumor-suppressor gene (TSG) densities contribute to amplification and deletion patterns of chromosomes[46,47], mutations in those genes could introduce a bias in the correlation analysis. Hence, we removed from the analysis the mutations within OGs and TSGs to assess whether they may inflate the correlations, which, in the end, were neither significantly decreasing nor increasing across all cancer types (Fig. 2k and Fig. 2l; = n.s.; paired two-sided Wilcoxon test) and indicated that cancer driver mutations do not bias our observations.

## Buffering events partially explain tissue-specific amplification patterns

After observing the association between the $\mu$ score and the amplification frequency, we wondered if this association could potentially account for the tissue specificity of amplification patterns. Previous research from Sack et al.[47] has addressed this question by performing a screen for proliferation modulators in breast and pancreatic cell lines using a barcoded genome-scale expression library. They calculated tissue-specific scores based on the density of modulators, as well as oncogenes (OGs) and tumor-suppressor genes (TSGs) along the genome. These studies have shown that those scores can partly explain the tissue-specificity of aneuploidy and CNAs[46,47]. Specifically, TSGs and genes that impede proliferation upon overexpression (STOP genes) tend to be frequently deleted, and, vice versa, OGs, essential and

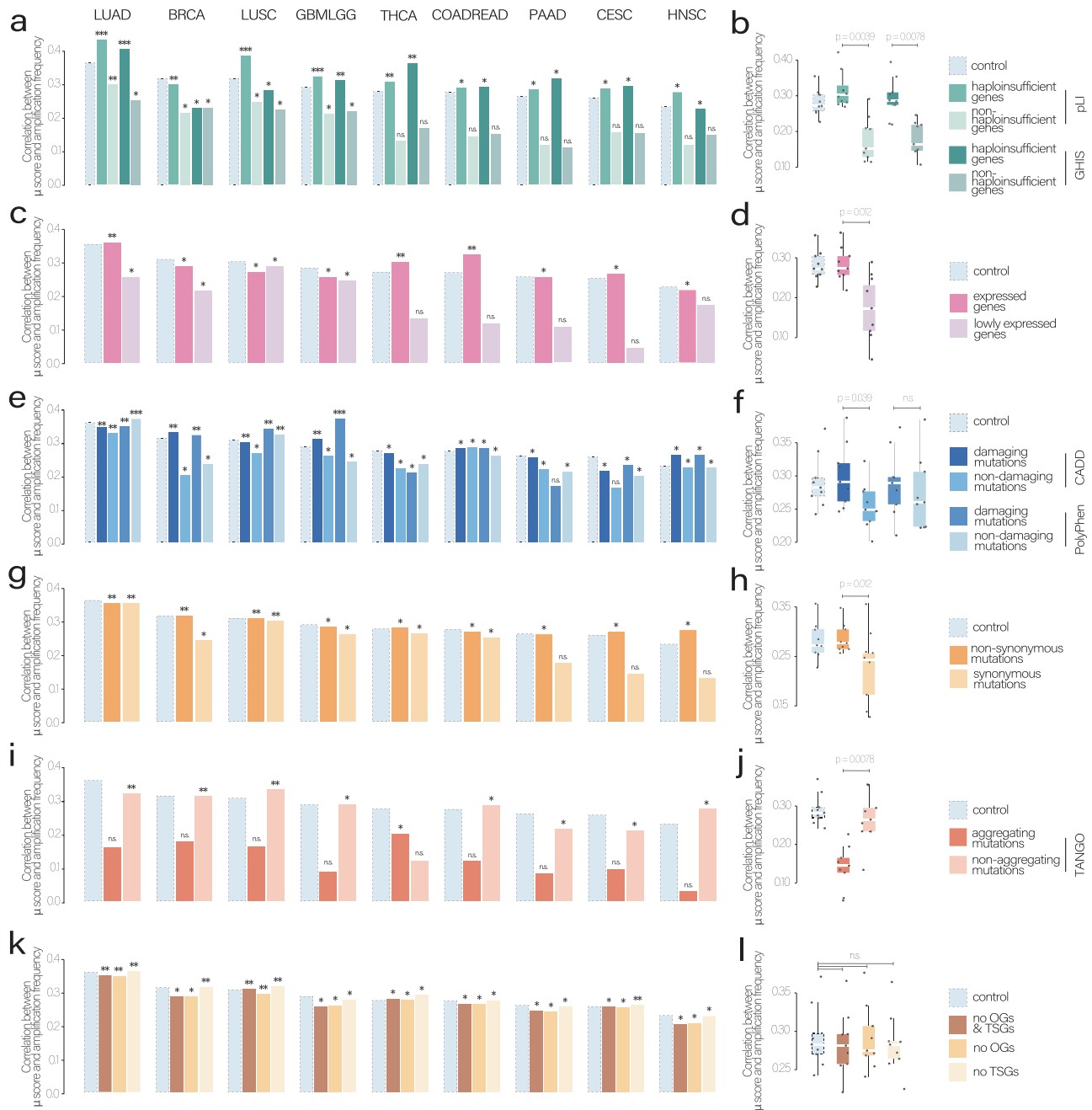

**Fig. 2 | Gene- and mutation-level properties modulate the buffering effect.** All correlations were compared to a control consisting of correlations between the mutation ($\mu$) score without any filtering and the amplification frequency **a** Spearman's correlations between amplifications and mutations within haploinsufficient (LOF-intolerant) genes and all amplifications and mutations within non-haploinsufficient (LOF-tolerant) genes using the probability of being LOF-intolerant (pLI) score and the genome-wide haploinsufficiency score (GHIS) independently. **b** Comparison of Spearman's correlations between haploinsufficient and non-haploinsufficient conditions. **c** Spearman's correlations between amplifications and mutations within expressed and non-expressed genes. **d** Comparison of Spearman's correlations between expressed and non-expressed conditions. **e** Spearman's correlations between amplifications and mutations classified using Polyphen or the combined annotation dependent depletion (CADD) scores as non-damaging or damaging. **f** Comparison of Spearman's correlations between damaging and non-damaging mutations using CADD and Polyphen. **g** Spearman's correlations between amplifications and synonymous and non-synonymous mutations. **h** Comparison of Spearman's correlations between synonymous and non-synonymous conditions. **i** Spearman's correlations between amplifications and aggregating and amplifications and non-aggregating mutations. **j** Comparison of

Spearman's correlations between aggregating and non-aggregating conditions. **k** Spearman's correlations between amplifications and mutations without both oncogenes (OG) and tumor-suppressor genes (TSG), as no OGs & TSGs, amplifications and mutations without TSG mutations (*no TSGs*), amplifications and mutations without OG mutations (*no OGs*). **l** Comparison of Spearman's correlations between no OGs & TSGs, no OGs and no TSGs conditions. For **a**, **c**, **e**, **g**, **i**, **k**, asterisks on top of each bar refer to Spearman's correlation *p*-values (*$p < 5e-02$, **$p < 5e-03$ and ***$p < 5e-04$). No multiple testing correction was performed. The number of genes and mutations in each category are indicated in Supplementary Dataset 4. For **b**, **d**, **f**, **h**, **j**, **l**, *p*-values are calculated using the paired two-sided Wilcoxon test, $n = 9$ tumor types in all cases. The box represents the 1st to 3rd quartile with the median marked by a horizontal line. LUAD lung adenocarcinoma, BRCA breast invasive carcinoma, LUSC lung squamous cell carcinoma, GBMLGG glioblastoma multiforme, brain lower grade glioma, THCA thyroid carcinoma, COADREAD colon adenocarcinoma and rectum adenocarcinoma, PAAD pancreatic adenocarcinoma, CESC cervical squamous cell carcinoma and endocervical carcinoma, HNSC head and neck squamous carcinoma. Source data are provided as a Source Data file.

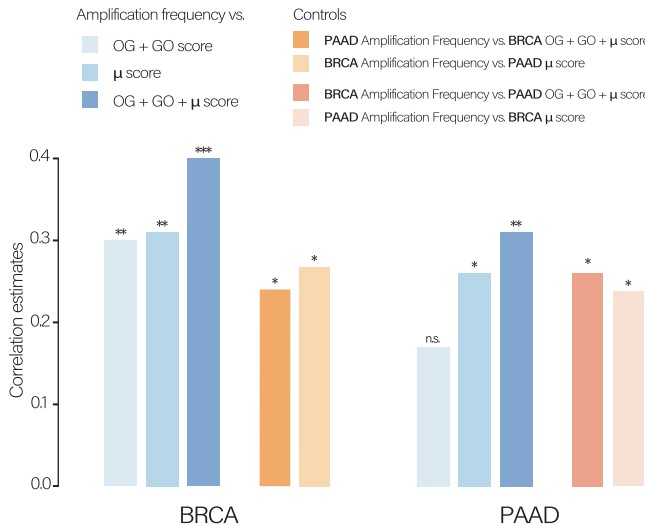

**Fig. 3 | The _μ_ score cooperates on predicting genomic amplification pattern.**
Breast invasive Carcinoma (BRCA) and Pancreatic adenocarcinoma (PAAD) correlations using different models: amplification frequency vs oncogene (OG) + proliferation-inducing gene (GO) scores; amplification frequency vs mutation ($\mu$) score; amplification frequency vs OG + GO + $\mu$ score. As controls, amplification frequency and OG + GO + $\mu$ score were swapped between BRCA and PAAD. Asterisks above bars refer to Spearman's correlations with *$p < 5e\text{-}02$, **$p < 5e\text{-}03$ and ***$p < 5e\text{-}04$. Source data are provided as a Source Data file.

proliferation-inducing genes (GO genes) tend to be enriched in recurrent amplifications in a tissue-specific manner. We reasoned that by incorporating different forces that shape the amplification landscape in tumors – the need to amplify proliferation drivers and protect cancer-essential genes –, we might be able to develop a more accurate model of amplification patterns. Based on the analyses from Davoli et al.[46] and Sack et al.[47], we designed a modified version of the OG and GO scores, respectively, by adapting the original version to a smaller segmentation length of $j = 36$ Mbp. We restricted our analysis to BRCA and PAAD since the initial screen for GO genes was performed only on human mammary epithelial cells (HMECs) and human pancreatic nestin-expressing epithelial cells (HPNEs)[47]. We then created a model that incorporates the OG and GO scores as well as our $\mu$ score, calculated using the same segmentation length ($j$).

We observed that by summing up the three factors (OG + GO + $\mu$ score), we increased the correlation with amplification frequency in both BRCA and PAAD (Fig. 3 and Fig. S4). As a control, we swapped both the amplification frequencies and the $\mu$ score between BRCA and PAAD, observing a drop in performance in both cases (Fig. 3), confirming the tissue specificity of the observed associations.

**Buffering events tend to precede mutations**
From a tumor evolution perspective, amplifications could exert a buffering effect, reducing the deleterious consequences of mutations, only if they temporally appear before somatic mutations. Thus, only if CNAs amplify wild-type regions, they can render the segments more tolerant to mutation accumulation. Therefore, we wondered if mutations are more likely to appear before or after the amplification events.

To assess the timing of somatic events along the tumor evolutionary trajectory, we utilized whole-genome sequencing (WGS) data within the Pancancer Analysis of Whole-Genomes (PCAWG) cohort[48]. WGS is the only type of assay that can determine timing information reliably. Using the CNAqc[49] tool for WGS we calculated the multiplicity ($m$) of each mutation, which measures the number of altered copies within a given segment's allele-specific CN and phases mutations and CNs over time (Fig. S5). When segments carrying early mutations are amplified, mutations are necessarily repeated within the segment

itself, resulting in a high $m$. Conversely, late mutations are not repeated within the amplifications, displaying a lower $m$. Intuitively, mutations with a closer $m$ to the major allele CN indicate an earlier mutation event from a timing perspective. This clock is therefore a simple approximation to determine the relative temporal ordering among mutations mapped to the same CN segment, especially for segments that do not carry a minor allele, such as LOH regions.

Using this principle and reasoning that $m$ can grow as much as the most-represented allele (i.e., the major allele), we classified the mutations with multiplicity equal to the major allele count as early events, as shown in Eq. (6) in the "Methods" section. Even when pooling mutations within different segment's specific CNs, our classification showed that overall half of mutations occur before the amplification events (50.2%); instead, when focusing on the mutations within a single allele-specific CN, they appear after the amplifications in most cases (Figs. 4a and S6a). The tendency of mutations of not being amplified highlights the reliability of our proposed evolutionary model where extra copies should be unmutated in order to compensate for further deleterious mutations in their segments.

Since the PCAWG data provides a complete profile of both coding and non-coding mutations, we tested whether those two types of mutation are differentially associated with predicted late or early classes. We observed a significant association between late mutations and coding mutations ($p = 9.6e\text{-}04$, Fisher's test). Interestingly, this association was particularly high for segment's allele-specific CN where the minor allele cannot buffer the deleterious mutations (i.e., x:0 and x:1 cases - such as 2:0, 3:0, 4:0, 5:0, and 2:1, 3:1, 4:1; Fig. 4a). In contrast, segment's allele-specific CNs where the minor allele provides at least two unmutated copies of the allele do not show a significant association between late mutations and coding mutations (i.e., x:2 and x:3 cases), except for the 3:3 case. In addition, we tested whether we observed a higher probability of having an early synonymous mutation than an early non-synonymous mutation, observing a statistically significant association (Fig. S6b; $p = 7.676e\text{-}10$; Fisher's exact test). Moreover, we computed the $\mu$ score using coding and non-coding mutations across PCAWG tumor types and found a stronger association between amplifications and coding than amplifications and non-coding mutations (Fig. 4b, c).

Finally, considering that our clock model based on mutation multiplicities cannot distinguish pre-amplification mutations in single copy ($m = 1$), from mutations after amplifications – a limitation that mildly impacts non-LOH regions – we opted to test an alternative clock model based on buffered ($\geq 2$ wild-type copy) and unbuffered ($< 2$ wild-type copy) mutations. We defined, based on the multiplicity, the degree of buffering in each cancer type and assessed how widespread this effect is, we defined buffered mutations as those with at least two unmutated copies given a segment's allele-specific CN, as shown in Eq. (7) in the "Methods" section. Our results showed that between about 75% and 95% of mutations in amplified segments were buffered (Fig. S6c, d). When considering the whole pool of mutations – in amplified, diploid, and deleted segments – the percentage of buffered mutations varied between about 25% to 50% of the total number of mutations, as shown in Fig. S6e.

**Protected genes are enriched in essential gene functions**
We wondered whether essential or cancer-specific gene functions are protected by amplifications while non-essential and cancer-unrelated gene classes tolerate the accumulation of mutations. Based on the amplification frequency and the $\mu$ score, defined on the same genomic scales, we calculated a protection index ($P_i$) for each protein-coding gene and the associated distribution percentile, calculated by permuting 10,000 times $P_i$ (see "Methods" section). Based on $P_i$ percentiles, we classified genes as protected and non-protected. Then, we performed Gene Ontology analysis on these two gene sets to better understand functions under protection. This analysis revealed that

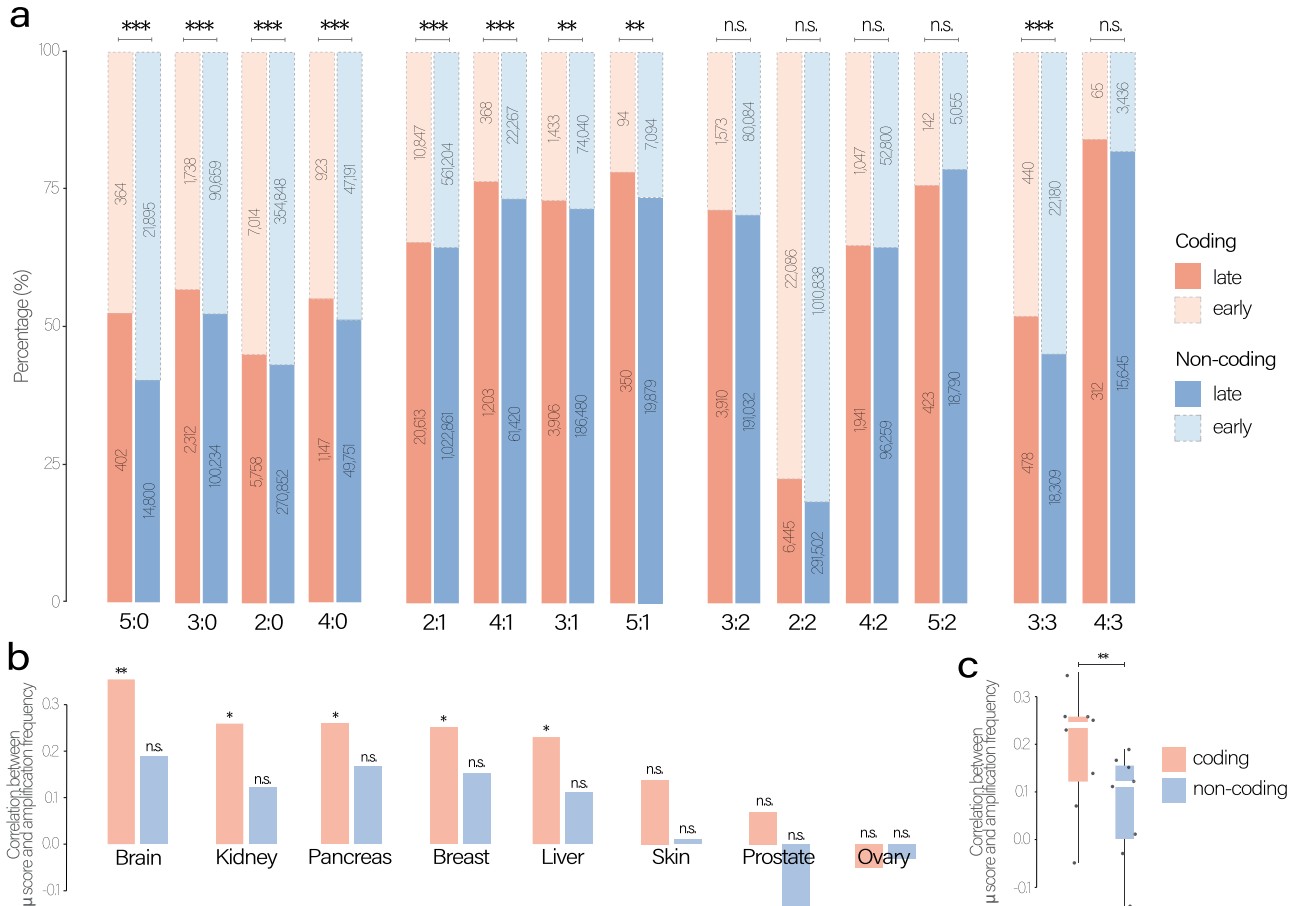

**Fig. 4 | Mutation-amplification relative timing. a** Proportion of early and late events between coding and non-coding mutations in each segment's allele-specific copy-number. The significance of the association between coding mutations being late events was determined using Fisher's exact test, with p-values shown on top of each bar pair (*$p < 5e-02$, **$p < 5e-03$ and ***$p < 5e-04$). The order of the segment's allele copy number is based on the magnitude of the minor allele, and the statistical associated p-value. The numbers within each bar represent the number of mutations within specific categories. **b** Spearman's correlations between amplifications and mutations within coding and non-coding regions. Spearman's correlation p-value are shown on top of each bar (*$p < 5e-02$, **$p < 5e-03$ and ***$p < 5e-04$). **c** Comparison of Spearman's correlation distributions between coding and non-coding conditions ($p = 0.015$; paired two-sided Wilcoxon test; $n = 8$ tumor types). The box represents the 1st to 3rd quartile with the median marked by a horizontal line. Source data are provided as a Source Data file.

unprotected gene sets tend to contain unessential, tissue-specific, and cancer-unrelated biological functions: most strongly enriched terms were "detection of chemical stimulus involved in sensory perception", "G-protein coupled receptor signaling pathway, coupled to cyclic nucleotide second messenger", "regulation of vasoconstriction", "regulation of cytosolic calcium ion concentration" and "G1 to G0 transition" ($q < 0.05$; Fig. 5a). Conversely, protected gene sets were enriched in essential cellular functions relevant for tumor cell homeostasis, such as "cellular macromolecule metabolic process", "phospholipid biosynthetic process", "protein transport", "mitochondrion organization" and "protein metabolic process" ($q < 0.05$; Fig. 5b).

To test if protected genes tend to be essential, we retrieved the gene essentiality score from the Achilles CRISPR screens[50,51] (the CRISPR gene effect). Then, we compared the score between protected and unprotected genes. The CRISPR gene effect distributions confirmed that protected genes do impact more cellular fitness compared to unprotected ones ($p = 8.61e-16$; Wilcoxon test - Fig. 5c). As a further validation, we assessed the expression of protected and unprotected genes across tumor types, observing that the protected gene set had an overall higher expression than the unprotected gene set (Fig. S9; $p < 2.22e-16$ in all tumor types; Wilcoxon test). We then compared the tissue-specific expression of both protected and unprotected gene sets, observing a higher tissue-specificity for the unprotected gene set, and vice versa, an overall high expression across tissues for the

protected gene set (Fig. 5d; $p = 3.21e-34$; two-sided Wilcoxon test). Moreover, we compared the overlap of CRISPR common essential and non-essential genes[51] within the two categories of protected and unprotected genes: we observed a statistical association within protected and common essential ($p = 6.12e-26$; Fisher's test). Protected, unprotected, common-essential and non-essential gene lists are provided in Supplementary Dataset 3.

## Discussion

In this work, we address the conundrum of how cancers could tolerate the presence of potentially deleterious mutations and why we do not see strong signals of avoiding mutations in cancer-essential genes. As a matter of fact, the majority of genes lack signals of negative selection in tumor genomes. However, mounting evidence suggests that many of those mutations are associated with a fitness cost per se[7–10,13,14].

Likewise, though somatic amplifications are frequently associated with a fitness cost – higher DNA content to carry and duplicate, increase of genome instability and protein imbalance[36,39,52–60] – they are known to play an important role in cancer progression by promoting the overexpression of oncogenes and other key genes that contribute to the development and maintenance of the tumor phenotype[46,47]. However, our study sheds light on the potential buffering role that amplifications may play in the context of passenger mutations.

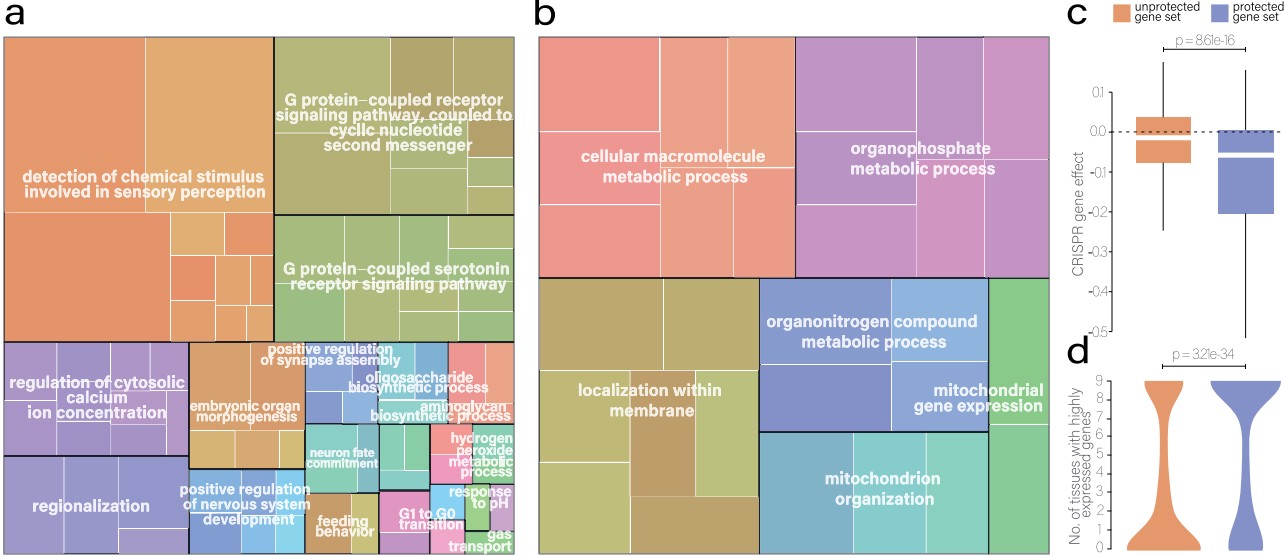

**Fig. 5 | Functions of protected and unprotected genes. a** Gene ontology enrichments of unprotected gene functions (ReViGO). **b** Gene ontology enrichments of protected gene functions (ReViGO). The full set of all gene ontology terms is shown in Fig. S7. **c** CRISPR gene effect for protected and unprotected genes. The *p*-value is determined by a two-sided Wilcoxon test (protected: *n* = 633, unprotected: *n* = 1140; boxplot with outliers in the supplement, Fig. S8). The box represents the 1st to 3rd quartile with the median marked by a horizontal line. **d** Number of protected and unprotected highly expressed genes across tumor types. The *p*-value is determined by a two-sided Wilcoxon test.

By analyzing genomic data from a large cohort of patients in TCGA and PCAWG, we found that amplifications can act as buffers that enable tumor genomes to tolerate the accumulation of deleterious passenger mutations. This buffering effect may help to reduce the fitness cost of accumulating deleterious mutations, allowing the tumor to maintain growth and survival (Fig. 6). This effect depends on the tumor-specific mutation burden. In asexual populations as tumors, cells irreversibly accumulate mutations and could potentially experience a decrease in fitness as a consequence. In fact, due to their high mutation rate, the ratchet-like accumulation of deleterious mutations would increase the likelihood of population extinction through mutational meltdown. The higher the mutation rate, the faster the population may potentially become extinct, unless some buffering effect comes into play. As shown in Fig. 1f, taking correlations between the amplification frequency and the *μ* score as a proxy for buffering, we showed that this effect is increasing with higher mutation load.

Our model implies that some amplifications might be selected to buffer mutations regardless of the presence of oncogenic alterations in the amplified region (Fig. 2k, l). This suggests that these amplifications may constitute a distinct class of events within the cancer genome, which goes beyond the classical categorization of genomic alteration into drivers or passengers.

Although the hypothesis that there is an intricate link between ploidy and relaxation of selection was already proposed before[4,5,28–32,61,62], we here provide several pieces of evidence that the observed amplification patterns in tumors (~90% of tumors are aneuploid[37,63] and even more holding focal CNAs[60]) linked to regional differences of mutation rates, moving away from the unique interpretation of amplifications as linked to increased evolutionary advantage and positive selection[64]. Indeed, many species adopt similar mechanisms by increasing genomic copy numbers to counteract the ratchet-like accumulation of deleterious mutations. Some examples of these mechanisms include horizontal gene transfer in bacteria[32] and polyploidy in amoebas[30] or yeasts[31]. The same role has been hypothesized for whole genome doubling in cancer[34]. Moreover, many studies proposed that genetic redundancy – the presence of multiple copies of genes having the same biological function – confers genetic

robustness towards most aberrations, where they have little or no effects because genes are functionally compensated by other genes[33,65].

Our results indicate that amplifications might act as buffering events mostly on a small to intermediate genomic scale (from 20 Mbp to 40 Mbp). We hypothesized that larger amplification lengths (from 50 Mbp to chromosome level) have too high fitness cost compared to the deleterious impact of mutations. In comparison, smaller correlation lengths (from gene level to 20 Mbp) would require more try and error leading to a scattered somatic amplification landscape in order to cover most cancer-essential genes. In other words, amplifications will be selected whenever the fitness cost of deleterious mutations outweighs the cost of the amplifications, resulting in a mutation-amplification-selection balance (Fig. 6).

We demonstrated that the proposed buffering effect strongly affects haploinsufficient genes and genes of specific cancer-essential functions. This may explain why, for example, Martincorena et al.[5] observed no detectable negative selection in haploinsufficient genes and negative selection against mutations in essential genes located in haploid regions, but not in diploid regions. Our findings will allow us to reverse our approach and use signatures of highly protected genes to identify potential cancer vulnerabilities and guide novel therapeutic intervention. It is worth noting that despite the clear tendency observed for most of the correlations across cancer types, in some specific situations, we do not observe the general trend. We would like to point out, however, that many of the gene and mutation properties (such as mutation impact or aggregation propensity) are predictions based on various features, partly with limitations in accuracy or being highly tissue specific.

Beyond potentially allowing for the identification of new cancer vulnerabilities in the future, another important point about our model is that it explains to a certain degree the tissue-specificity of amplification patterns. Furthermore, despite being random events, evolutionary pressures might select amplifications that better satisfy cancer genome needs, e.g., the buffering of deleterious mutations. Of course, further studies about mutation-amplification-selection balance are needed to better elucidate and even predict the amplifications that

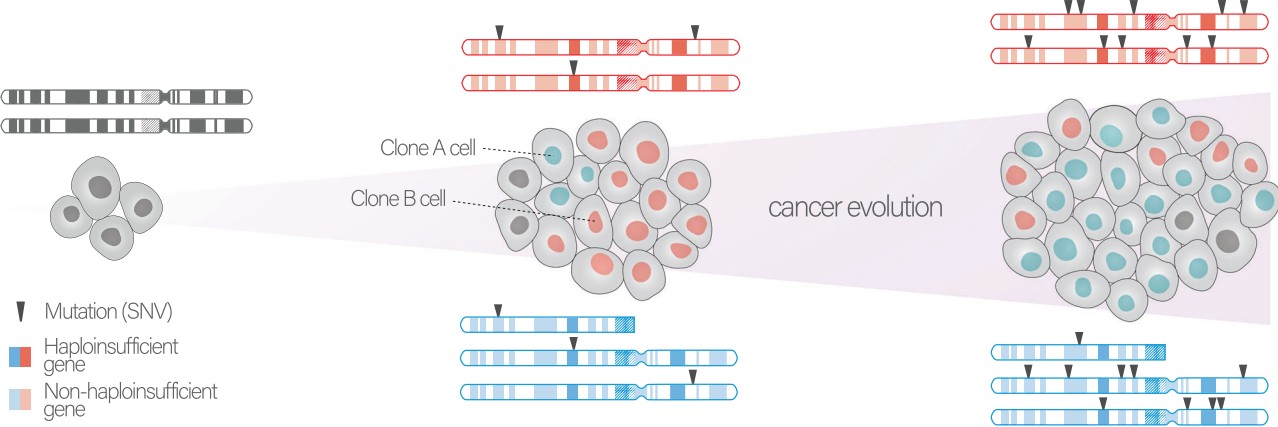

**Fig. 6 | Conceptual overview of the proposed evolutionary model.** Clone A remains diploid, while Clone B undergoes the amplification of a chromosome segment. Even though focal/arm-CN amplifications initially provokes a fitness loss compared to clone A, clone B fitness subsequently acquires a genetic advantage compared to clone A due to the better mutation buffering; indeed, in this scenario (clone B), two wild-type copies of amplified genes have been maintained, this effect is even more pronounced for haploinsufficient genes. Cell representations were adapted from "Icon Pack - Cell Biology" by BioRender.com, retrieved from https://app.biorender.com/biorender-templates/.

confer robustness to mutation accumulation and act as genomic buffering events during cancer evolution.

## Methods
### Unprocessed data download sources
Main data used for the analysis were CNAs, SNVs and clinical metadata from 8,690 solid primary tumors belonging to 23 TCGA projects: LUSC (Lung Squamous cell Carcinoma, $n = 477$), LUAD (Lung Adenocarcinoma, $n = 509$), COADREAD (Colon Adenocarcinoma and Rectum Adenocarcinoma, $n = 537$), CESC (Cervical Squamous cell Carcinoma and Endocervical Carcinoma, $n = 273$), BRCA (Breast invasive Carcinoma, $n = 996$), SKCM (Skin Cutaneous Melanoma, $n = 104$), OV (Ovarian Serous Cystadenocarcinoma, $n = 387$), UCS (Uterine Carcinosarcoma, $n = 56$), LIHC (Liver Hepatocellular Carcinoma, $n = 355$), HNSC (Head and Neck Squamous Carcinoma, $n = 496$), PRAD (Prostate Adenocarcinoma, $n = 477$), THCA (Thyroid carcinoma, $n = 479$), PCPG (Pheochromocytoma and Paraganglioma, $n = 159$), ESCA (Esophageal carcinoma, $n = 435$), STAD (Stomach adenocarcinoma, $n = 435$), GBMLGG (Glioblastoma multiforme, Brain Lower Grade Glioma, $n = 842$), KIRC (Kidney renal clear cell carcinoma, $n = 362$), KIRP (Kidney renal papillary cell carcinoma, $n = 278$), PAAD (Pancreatic adenocarcinoma, $n = 176$), TGCT (Testicular Germ Cell Tumors, $n = 144$), MESO (Mesothelioma, $n = 77$), SARC (Sarcoma, $n = 233$), and BLCA (Bladder Urothelial Carcinoma, $n = 403$). CNAs (mapped to the hg19 genome) and clinical metadata were downloaded from Firebrowse[66]. Preprocessed mutation data were downloaded from GDC data collection[67].

### Amplification/deletion frequencies and *μ* score calculation
Chromosomes were segmented with fixed segment length (*j*) from 1,000,000 bp, 50,000,000 bp (using a step width of 1,000,000 bp), arm to chromosomes and, within each segment (*i*), amplification frequency and mutation score (*μ*) were computed. These frequencies were calculated within each cohort of patients belonging to the same TCGA-project. A graphic illustration of the method is in Fig. S1.

Segments (*i*) were classified as copy-neutral ($pt^N_{i,j}$), amplified or deleted (or $pt^A_{i,j}$, $pt^D_{i,j}$ respectively) based on DNACopy[68] R-package parameter *segment_mean* as described in Bioinformatics Pipeline from GDC[69] for CNAs, using ±0.2 as cutoff. The amplification frequency was calculated as the number of patients with an amplified segment ($\#pt^A_{i,j}$) over the total number of patients ($\#pt^D_{i,j} + \#pt^A_{i,j} + \#pt^N_{i,j}$) within the analyzed TCGA-project.

Therefore, for each segment (*i*) with specific segment length (*j*):

$$Amplification\ frequency_{i,j} = \frac{\#pt^A_{i,j}}{\#pt^D_{i,j} + \#pt^A_{i,j} + \#pt^N_{i,j}} \quad (1)$$

The deletion frequency was calculated analogously:

$$Deletion\ frequency_{i,j} = \frac{\#pt^D_{i,j}}{\#pt^D_{i,j} + \#pt^A_{i,j} + \#pt^N_{i,j}} \quad (2)$$

Each CNA event was classified into chromosomal – if the event covers at least the 90% of the chromosome length –, chromosome-arm – if it covers more or equal than 50% of the arm length – and focal – if it covers less than 50% of arm length. Thus, chromosomal gain or loss frequencies were calculated according to Eqs. (1) and (2) using chromosomal or chromosome-arm gain or loss events, while segment gain or loss frequencies were calculated according to Eqs. (1) and (2) using all CNA events (chromosomal, chromosome-arm and focal) at the selected segment length (*j*).

Then, only somatic protein-coding mutations were used for the analysis (excluding both X and Y chromosomes). Mutations were counted only within copy-neutral segments ($\sigma_{pt^N_{i,j}}$) – that is genomic regions that have the same copy number in a given sample segment as compared to the reference genome. In other words, copy-neutral segments are regions that do not show any CNAs, i.e., they are neutral in their copy number state as compared to the dominant ploidy of the sample. After that, the extracted number of mutations ($\sigma_{pt^N_{i,j}}$) is normalized over the number of non-amplified patients within the segment ($\#pt^N_{i,j}$) and the number of protein-coding nucleotides within the segment ($bp^{coding}_{i,j}$) to obtain the mutation frequency (log10 is applied).

For each segment *i* with specific segment length *j*:

$$\mu_{i,j}\ score = \log_{10} \frac{\sum_{pt^N} \sigma_{pt^N_{i,j}}}{bp^{coding}_{i,j} \cdot \#pt^N_{i,j}} \quad (3)$$

To maximize the fidelity of mutation frequency estimation, mutations within uncovered region from SNP6.0 microarray (TCGA CNA pipelines[69]) were discarded from the analysis.

### Cancer correlation analysis
Segment correlations were performed using R-base functions calculating regression coefficients and associated p-values between each

TCGA project-specific amplification/deletion frequencies and the $\mu$ score using Spearman's method. Correlations were performed for each segmentation length ($j$ = gene, from 1 Mbp to 50 Mbp, chromosome-arm and chromosome). Only significant cancer types (Spearman's $\rho < 0.05$) were considered for further analysis (darker bars in Fig. 1c).

To be consistent with the single TCGA-project correlation analysis (previous paragraph), overall $\mu$ score for each TCGA project was calculated as the mean of the not-log transformed $\mu$ score, then log10 transformed. Accordingly, the overall amplification/deletion estimates were calculated as the mean frequency of amplification/deletion.

## Gene properties and enrichment analysis

Haploinsufficiency was assessed using the measure of tolerance upon LOF: the probability of being LOF intolerant (pLI) score[40]. The calculation is based on the assumption that haploinsufficient genes are less tolerant to variants. For each gene, the pLI method calculates the Z-scores that indicate the deviation from the observed counts from the expected number of synonymous and missense mutations (a positive Z-score indicates an increased intolerance to variants, and vice versa negative Z-scores indicate that genes have more variants than expected). Using this pLI score, mutations were divided based on those that occur in haploinsufficient and non-haploinsufficient genes, performing the correlation analysis separately. The cutoff was chosen to get a consistent and comparable number of genes/mutations in the defined categories. Hence, genes with pLI > 0.2 were considered as haploinsufficient, while with pLI ≤ 0.2 were considered as non-haploinsufficient (mutations within genes without calculated pLI score were discarded from the analysis). Furthermore, the $\mu$ score was calculated only using the mutations within genes of the two categories. In addition, we distinguished haploinsufficient and non-haploinsufficient genes using a different predictor, the genome-wide haploinsufficiency score (GHIS) score[41]. This algorithm relies on a support vector machine that integrates known features of haploinsufficient genes (co-expression, coding sequence length, evolutionary constraints, as well as genetic variants), to calculate a genome-wide haploinsufficiency score. The cutoff was chosen to get a consistent and comparable number of genes/mutations in the defined categories. Hence, genes with GHIS > 0.5 were considered as haploinsufficient, while GHIS ≤ 0.5 were considered as non-haploinsufficient (mutations within genes without calculated GHIS score were discarded from the analysis). Furthermore, the $\mu$ score was calculated only using the mutations within genes of the two categories.

Gene expression information (transcripts per million) was retrieved from TCGA, and the median value was calculated across patients within each tumor type. All the genes with a median expression of zero were defined as non-expressed.

Cancer genes (OGs and TSGs) were extracted from UniProt[70], Cosmic[71], TUSON Explorer[46] and Intogen[72].

Gene ontology-enrichment analysis was performed using *ClusterProfiler::enrichGO()*[73] function. ReVIGO[74] was further used to refine and reduce the redundant enriched Gene ontology terms. As a measure of essentiality, the genes within protected and unprotected sets were compared using the scores derived from CRISPR-knock-out screens, called CRISPR gene effect[50,51] scores.

## Mutation properties

Mutations were classified as strongly or lowly damaging. Pre-calculated scores were downloaded from OncoVar[75]. CADD score cutoffs were above 3.5 and below 3.5 for high and low impact, respectively; PolyPhen score cutoffs were above 0.6 and below 0.3 for high and low impact, respectively.

Mutations were furthermore classified as aggregation-causing and non-aggregation-causing based on the aggregation score and fold change, calculated as the mutated over the wild-type aggregation score. TANGO was used to predict protein aggregation propensity

upon the introduction of the mutation in the protein sequence and it was run for every wild-type and mutated protein. The computation considered only mutations which cause a change in the amino acid composition of the protein. Mutations with the aggregation score above 5000 units and the fold change (FC) above 1 were classified as aggregation-causing mutations, while all other mutations were considered non-aggregation-causing.

All cutoffs - CADD[43], Polyphen[42] and TANGO[45] scores - were chosen to get a consistent and comparable number of genes or mutations in the defined categories.

## OG and GO scores calculation

OG and GO score were calculated as in Davoli et al.[46]. and Sack et al.[47], generalized in order to have a score for each segment $i$ with a defined length $j$ = 36 Mbp (and not only for entire chromosome or chromosome-arms):

$$OG\,score_{i,j} = \frac{\#OG_{i,j}}{\#genes_{i,j}} \qquad (4)$$

$$GO\,score_{i,j} = \frac{\#GO_{i,j}}{\#genes_{i,j}} \qquad (5)$$

Those scores represent the relative density of OGs and GOs within each segment ($i, j$). To calculate the *OG score* we selected the first 250 OGs, decreasingly ordered by p-value, according to TUSON (TUmor Suppressor and ONcogene) Explorer[46]. Analogously, to calculate the *GO score* we used the results of the overexpression genetic screens of proliferation-inducing genes from Sack et al.[47], using the same filtering criteria described in the original paper. For this analysis, the $\mu$ score was scaled between 0 and 1 in order to have comparable data ranges.

## Mutation timing using CNAqc

The CNAqc[49] R package was used to perform the analysis via its *advanced_phasing()* function that computes mutation multiplicity ($m$) for all possible input allele-specific CN segments. For the timing analysis we used PCAWG Whole-Genome sequencing data released with the CNAqc, which is available at https://zenodo.org/record/6410935#.YrxMxexBweU. CNAqc infers clonality and mutation $m$ from the variant allele frequency spectrum, allele-specific segments, and tumor purity. Technically, $m$ represents the number of copies of a mutation, if it sits on a tumor genome segment with $n$ total copies (i.e., $n$ total copies of a major and minor allele in the tumor). According to the early or late classification, defined in (6), we defined as early the mutations that appeared with $m$ equal to the major allele, and as late the mutations that appeared with $m$ lower than the major allele. In the same way, we defined mutations as buffered or non-buffered based on $m$. Given a segment's allele-specific copy-number, we defined as buffered the mutations in segments with at least two unmutated copies as described as in Eq. (7).

$$CN_{major\,allele} \begin{cases} = m \rightarrow late \\ > m \rightarrow early \end{cases} \qquad (6)$$

$$\left(CN_{major\,allele} + CN_{minor\,allele}\right) - m \begin{cases} \geq 2 \rightarrow buffered \\ < 2 \rightarrow non\,buffered \end{cases} \qquad (7)$$

## Protected and unprotected cancer gene functions

The $\mu$ score and the amplification frequency were assessed for each protein-coding gene within the cohort of analyzed cancer types as described in previous sections. Non-expressed genes (TPM = 0) were removed from the calculation. Genes were classified in protected and

unprotected based on protection index ($P_i$), calculated as: $P_{i,gene} = Amplification\ frequency - \mu\ score$, followed by conducting 10,000 permutations for each gene, swapping amplification frequency and the $\mu$ score. Genes above the 94th percentile of $P_i$ were classified as protected, and below the 6th percentile as unprotected. The analysis was performed independently for each cancer type. To determine whether genes were protected or unprotected, we considered only those present in at least three out of the nine analyzed cancer types.

## Statistics and reproducibility
No statistical method was used to predetermine sample size. No data were excluded from the analyses.

## Reporting summary
Further information on research design is available in the Nature Portfolio Reporting Summary linked to this article.

## Data availability
TCGA genomic and transcriptomic data are available from NCI Genomic Data Commons (GDC) [https://portal.gdc.cancer.gov/] and were collected from FireBrowse [www.firebrowse.org]. The processed PCAWG data used in this study were collected from CNAqc repository available in Zenodo [https://zenodo.org/record/6410935#.YrxMxexBweU][76]. The CRISPR score and cancer cell line information data used in this study are available in DepMap project [https://depmap.org/portal/]. The mutation impact predictor scores (CADD and PolyPhen) data are pre-calculated and available on OncoVar [https://oncovar.tania.wang:5443/welcome/download]. The haploinsufficiency score data are available on [https://static-content.springer.com/esm/art%3A10.1038%2Fnature19057/MediaObjects/41586_2016_BFnature19057_MOESM241_ESM.zip] (pLI) and [https://academic.oup.com/nar/article/43/15/e101/2414292?login=false#supplementary-data] (GHIS). The cancer gene definitions used in this study are available in the Uniprot database [https://www.uniprot.org/], in the COSMIC database [https://cancer.sanger.ac.uk/census], in IntOGene [https://www.intogen.org/download] and in TUSON Explorer [https://www.cell.com/cms/10.1016/j.cell.2013.10.011/attachment/3204ad4e-e9f5-44c8-9ae5-69b792d28490/mmc4.zip].
The overexpression genetic screens of proliferation-inducing genes are available on [https://www.cell.com/cms/10.1016/j.cell.2018.02.037/attachment/1fe998bd-09f9-4f50-832c-eccf25cfc7f9/mmc2.xlsx]. The data generated in this study have been deposited in Zenodo [https://doi.org/10.5281/zenodo.7079304][77]. Source data are provided with this paper.

## Code availability
The analysis scripts were developed using R (version 4.2.1) and Python (version 3.9.12). The code generated in this study have been deposited in GitGub [https://github.com/fabio-alfieri/mutation_compensation][78].

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

## Acknowledgements

The work leading to this manuscript was supported by Fondazione AIRC, grant reference number MFAG21791 and partially supported by the

Italian Ministry of Health with Ricerca Corrente and 5×1000 funds. G.C. was supported by Fondazione AIRC, grant reference number MFAG24913. F.A. is a PhD student within the European School of Molecular Medicine (SEMM). The results shown here are in whole or part based upon data generated by the TCGA Research Network: https://www.cancer.gov/tcga.

## Author contributions

F.A. performed the statistical and computational work. G.C. gives contributions to the reconstruction of tumor evolution of the analyzed PCAWG samples. M.S. conceived and supervised the work. F.A. and M.S. wrote the manuscript with contributions of G.C. All the authors read and approved the final manuscript.

## Competing interests

The authors declare no competing interests.
