## [Peer review file · Nature Communications]

REVIEWER COMMENTS

Reviewer #2 (Remarks to the Author): Expert in computational cancer genomics and evolution

The manuscript of Alfieri et al. describes a correlation analysis between copy number amplification (CNA) and somatic mutation frequencies in cancer genomics data from The Cancer Genome Atlas (TCGA). Based on these correlations they suggest that CNAs are protective (or compensatory?) mechanisms for the deleterious effects of somatic mutations in (haploinsufficient) essential genes. This could explain why signals of negative selection have been found largely absent in cancer.

The manuscript is overall well-written and easy to follow. The starting hypothesis is very intriguing and the authors made a substantial effort to provide several analyses in support of this hypothesis.

My main criticism is the purely correlation nature of the study. Obviously, correlations don't mean causation and in my experience are rarely what they seem in these kinds of genomic analyses. In this regard I was lacking some orthogonal (ideally wet lab) validation of the authors' findings. I was also surprised that the authors did not discuss the pure correlational nature as a limitation of their study.

My main concerns are the putative confounding effects of (1) genomic instability and/or (2) the trivial correlation between mutation rates and copy number variations (CNVs).

The authors used one approach to exclude that genomic instability is a confounder, by focussing on copy number deletions (CNDs). While they found a (non-significant) similar correlation between mutations and deletions (fig. S2), when focussing on individual cancer types (fig. 1D) they found a clear difference between CNAs and CNDs. It seems though that the correlation becomes negative for deletions? Could the authors comment on this? Doesn't this simply indicate that less mutations are detected in (hemizyously) deleted areas (only one chromosome segment) versus amplified areas (multiple segments). The authors tried to correct for this by only using mutations from copy number neutral segments, but as the classification as CNN/CNA/deletion depends on arbitrary cut-offs I'm concerned that this analysis indicates that the problem still remains. Given these findings, I strongly disagree with the rather absolute statement that "these results discarded that possible confounding explanation ..." (line 120-121).

One additional negative control analysis that could be considered is a focus on mutations in non-expressed genes (like the other analyses shown in fig. 2). Correlations should be largely absent here if the authors' interpretation is correct. Additionally, the analyses shown in fig. 2 is now univariate only. How about interaction between these variables? E.g., non-damaging mutations in haploinsufficient genes?

Could the authors comment why the correlation was only significant in 9 cancer types (fig. 1C)? And why these exact cancer types? I find it remarkable that all 9 cancer types have a high mutational burden and/or TCGA sample size. Additionally, I would suggest that all downstream analyses (e.g., fig 1D, S3, ...) are performed on the complete set of cancer types and added as supplementary figures.

Based on their results, the authors interpret the amplifications as a protective mechanism for the deleterious effect of somatic mutations occurring later during tumor evolution (nicely summarized in fig. 6). However, on several occasions in the manuscript, they call the CNAs “compensatory events”. Isn’t this contradictory as this suggests CNAs compensate for mutations and hence occur later in tumor evolution? I found this very confusing. Additionally, assuming the CNAs occur first, could the authors speculate on the mechanism explaining the occurrence of these CNAs? Why would a cancer cell gain a CNA in the first place as it might cause a short-term fitness disadvantage as the authors indicate in fig 6?

Regarding the enrichment analysis shown in Fig. 5. I found it hard to interpret this figure and I don’t see the enrichments that are discussed in lines 254-259. The semantic spacing doesn’t improve interpretation here, so I would advise a simpler, more intuitive plot (e.g., barplots with ranked enrichment p values).

I found the methodology sometimes hard to understand, often lacking details necessary to understand how the analyses were performed exactly. Some examples: how is μ score calculated (explained in fig. S1 but more intuitive explanation needed in methodology), how are haplo(in)sufficient essential genes defined (+ how many are classified in each group), what is the OG/GO score (only possible to understand for a reader when reading the citations at the moment), ...

Some minor comments:

- Fig. S1: color legend is lacking
- Fig. S2: “mutation score” in x axis. Is this the same as “ μ score”?
- Fig. 1e: “correlation length” should be “segmentation length”?
- Many figures/analyses often lack “n=” details, e.g., how many haplo(in)sufficient, aggregating ... genes were found in fig. 2?
- Line 130: Where do I see p values in plot shown in fig. 1e. Only correlation coefficients are shown? Relatedly, several plots show (Pearson?) correlation coefficients, not just “correlations” as indicated in axes.
- I found fig. S3 more useful than main fig. 2. The authors could consider swapping main and supplement here.

- The added value of the analysis shown in fig. 3 is limited. Why is this analysis only performed on 2 cancer types?

- Added value of non-coding gene analysis in fig. 4 unclear. Which analysis does the Fisher test refer to? In general, I found the analysis in this figure hard to understand.

- Line 264: would add wilcox p value on plot

- Line 435: above the 10th percentile. 90th percentile?

In conclusion, while I like the starting hypothesis of this extended analyses, additional evidence is required that demonstrates convincingly that the observed correlation is not biased, before this manuscript is publishable in Nature Communications.

Reviewer #3 (Remarks to the Author): Expert in bioinformatics and computational cancer genomics

Fabio Alfieri and colleagues suggested in the manuscript that copy number amplifications in cancer genomes could be used as a way to buffer damaging variants by creating multiple unmutated copies as backups. This idea has been proposed before, such as in Martincorena et al. Cell 2017, but here the authors provided several new lines of evidence to support this hypothesis: (1) mutation rate is positively correlated with amplification rate; (2) damaging mutations, haploinsufficient and essential genes tend to be more protected by amplifications; (3) amplifications tend to appear earlier than mutations during somatic evolution.

Overall, I found that this manuscript is easy to follow and quite interesting, and the majority of the results are supported by solid analysis. The topic of this paper would be found useful and important by readers from the cancer genomes community. However, I think there are a few major and minor points needing to be addressed to further improve the manuscript.

Major points:

1. As the authors mentioned, the lack of negative selection was found by the dN/dS method. Have the authors tried to separate synonymous vs non-synonymous mutations when computing the association between the mu score and the amplification frequency in Fig. 1? Ideally, there should be stronger associations for non-synonymous mutations as synonymous mutations do not need the buffer. I would suggest the authors use synonymous mutations as their controls and compute mu_synonymous and

mu_nonsynonymous for their analysis. This was only partially answered in Fig. 2b. It's also related to Fig. 2a, I am wondering if the differential correlations for haploinsufficient genes when only predicted LOF mutations were used.

2. Similarly, the authors used PCAWG whole-genome sequencing data, so have the authors tried to compute mu scores for different categories of non-coding regions (promoters, enhancers, lncRNA and intergenic regions, etc.)? It would be interesting to see how the buffer effect changes among coding and non-coding elements.

3. It has been shown in multiple previous studies that genomic features such as chromatin states, replication timing, GC content etc. have an impact on somatic mutation rate due to differential DNA repair and/or damage rates. If these features also affect amplification rates, then the correlation at the segment level can also be explained. Could the authors rule out this possibility?

4. Many correlations observed by the authors were at the cohort or cancer type level (Fig. 1), but if copy number gains actually have the buffering function, they must co-occur with damaging mutations at the same genomic segment in the same individual. One would expect that, for example, the number of bases amplified should be positively correlated with the number of mutations across patients. I am wondering if the authors have observed this type of correlation as well. Moreover, if the buffering hypothesis is true, direct speculation should be that at least some amplified segments should contain heterozygous damaging variants (synonymous mutations can be used as control) at the same individual. Could the authors find direct evidence for this and find out how widespread this phenomenon is?

5. The authors found an optimal segment length for amplifications, which is quite interesting. It looks like this optimal segment length is the optimal parameter for calculating the correlation, but not the optimal length of CNAs tumors can have. In fact, have the authors checked the distribution of CNA length (should be available from WGS data) and how frequent is the 36 Mbp length? Moreover, can the authors speculate the biological meaning of such an optimal length?

6. In lines 136-137, the authors implied that mutation accumulation could drive genome amplification. However, temporally, mutations should appear after the genome amplification given the buffering hypothesis. Could the authors elaborate a bit more on how biologically this can happen?

7. In Fig. 4, the analysis of mutation-amplification relative timing, it looks like the difference between coding and non-coding mutations is small. As in point 1, what if the authors separate synonymous and nonsynonymous mutations? Would there be a larger difference?

Minor points:

1. In line 99, the authors mentioned that "the obvious confounder that the more copies of a segment exist the more mutations the segment could acquire". I am wondering if the authors have also calculated mu scores for amplified or deleted regions.

2. The mu score measurement of mutation frequency is crucial to many conclusions in the paper. In many places, such as Fig.1b, the authors used the mean mu score. Since in many tumor types, there are hypermutated individuals (mutation rate >30/Mb), the mean mu score may be skewed by them. I am wondering if the authors have filtered out hypermutated samples.

3. In Fig.1d, it looks like deletions have negative correlations with the μ score at the segment level. Do the authors have an explanation for this?
4. In the section below line 247, could the authors provide the number of protected and unprotected genes, common essential and nonessential genes, as well as their overlaps?
5. In line 255, the authors mentioned “most strongly enriched terms were “olfactory receptors”, “sensory perception of taste” and “heart cell development and morphogenesis””, but not all terms are obvious in Fig. 5a.
6. In Methods, the authors stated that “the genes above the 10th upper percentile of P_i were classified as protected, and below the 5th percentile as unprotected”. Why use these asymmetrical cut-offs (10th and 5th)?

Reviewer #4 (Remarks to the Author): Expert in cancer evolution and mathematical modelling

Alfieri et al. present an interesting study of the role of somatic copy number amplification in cancer, particularly in terms of its contribution to the protection of functional genes from mutation. While the question of whether gene duplication is beneficial against mutation is certainly not new (e.g. PMID: 7768451, 17954278, 32139907), these authors provide unprecedented genomic evidence in favour of the hypothesis that amplification can be advantageous to cancer cells not only by amplifying oncogenes, but also by generating additional intact copies of important genes.

The results of this study are of relevance to the cancer research field, especially as they highlight one mechanism by which tumour cells may become tolerant of the effects of their own genomic instability – although such a mechanism is not as radically novel as the authors seem to suggest. The methodology of the study also seems adequate in general. I am more concerned, however, by the article's notable lack of clarity and preciseness with respect to a number of its central points. As it is, the manuscript seems rather ambiguous in expressing the authors' interpretations of their findings. Specifically, it is often unclear whether the authors are defending a surprising sort of teleology, whereby gene amplification is somewhat purposefully produced with the aim of protecting important genes against future mutations. It is really only near the end of the article that one becomes reasonably sure of understanding the authors' argument – which is fortunately not the one I have just outlined. Moreover, some of the most central concepts of the paper (the “ μ score” and “mutation-amplification-selection balance”) are ill-defined or not defined at all, with the authors seemingly taking their meaning for granted. Therefore, I think it is especially important that these issues of intelligibility be addressed by the authors, to avoid the article from conveying misleadingly inaccurate arguments.

Below I list a series of major and minor points which I believe should be addressed before publication. In most cases, I cite the relevant sentence in the manuscript before my comments.

MAJOR COMMENTS

1.

The concept of the " μ score", which is central to the paper and is mentioned repeatedly since the Introduction, is surprisingly not defined anywhere in the main text. Moreover, its alternative name of "mutation score" is only used in the legend to Figure 1 and the Methods, compounding the confusion. It is even difficult to find the definition and formula for this score in the Methods. Such a lack of detail about one of the most important concepts in the paper is quite concerning. The authors should address this problem by taking care to:

- Introduce the concept of the μ score at a suitable early point in the main text.
- Make it clear in the Methods that Eq. 2 refers to the μ score.
- Elaborate their rationale for the μ score formula. Is this an original formula? If so, why was it necessary to develop it, and what is the strength of this particular measure? If not, what is the original source of this score?

2.

I have encountered several instances where the authors' phrasing is notably ambiguous, as it suggests that they might believe copy number amplification events to be explicitly "aimed" at protecting these regions from mutation, or even that they are triggered by mutation itself. After reading the entire paper, it is clearer to me that the authors believe in the (likely correct) view that these amplifications are random events which precede the mutations they are thought to protect against, and that it is due to selection during tumour evolution that these amplifications end up being overrepresented around certain types of genes. However, while one is in the process of reading the paper, the theoretical framework of the authors is not at all clear, and nearly every discussion of their results can easily be construed as a defense of the view that cancer genomes are deploying these amplifications in a strategic fashion in response to mutation. I would therefore encourage the authors to rework the text very carefully in order to avoid these unintended ambiguities. Below I list some such cases of ambiguous phrasing.

- p1: "undergo frequent somatic amplifications of their wild-type form to protect them from deleterious mutations".

This phrasing implies that protection of these genes is the cause of the amplification, rather than its consequence.

- p4: "we demonstrated that cancer-relevant regions undergoing high rates of damaging mutations tend to be amplified".

Is this phrasing accurate? Presumably tumour-suppressor genes (which are also relevant to cancer) do not tend to be amplified. More importantly, the sentence may be read as arguing that these regions are duplicated more frequently (even in the absence of selection), rather than their amplification being selected more frequently.

- p4: "the hypothesis that cancer evolution selects somatic amplification events to mitigate the irreversible accumulation of deleterious mutations by creating extra safe copies of highly mutated regions".

I think this would be better phrased as "somatic amplification events are selected in cancer evolution, as they mitigate...". The sentence as it is seems to suggest that cancer evolution intentionally produces these events beforehand to mitigate the future accumulation of mutations. I also think that "intact copies of genes found in highly mutated regions" would be more correct than "safe copies of highly mutated regions", as the term "safe" is not ill-defined in this context.

- p6: "this suggests that amplifications may be driven by mutation accumulation and not vice versa".

This may be read as implying a causal relationship ("driven") between mutation and amplification, which would of course contradict the finding that amplifications tend to precede mutations in those regions.

- p12: "A lot of species selects CN increase as a rescue mechanism to escape from the ratchet-like accumulation of deleterious mutations: horizontal gene transfer in bacteria, polyploidy in amoebas and whole genome doubling in cancer, which are all mechanisms aiming to the creation of safe wild-type copies of genes".

Besides needing some polishing, this sentence should be rephrased to avoid the message that amplifications have the "aim" of creating back-up copies of genes.

p9: "From a tumor evolution perspective, amplifications could exert a compensatory effect, buffering the deleterious consequences of mutations, only if they temporally appear before somatic mutations. Thus, only if CNAs amplify wild-type regions, they can render the segments more tolerant to mutation accumulation".

In relation to the previous point, it would be very helpful to introduce this idea in the Introduction. As explained above, it is currently not clear (until this point in the text) whether the hypothesis being proposed is that amplifications occur in wildtype regions and are later selected for their buffering effect, or alternatively that amplifications occur specifically in response to mutation.

4.

p2: "However, in many systems – including cancer cells – the majority of mutations have been observed to have a deleterious role on cellular fitness ... Specifically, among the different mutation types, more than 75% are non-synonymous ...".

It should be made very clear that this refers to mutations in coding-protein regions. This is an important point when speaking of the neutrality or deleteriousness of passenger mutations, as the vast majority of passenger mutations genome-wide are in noncoding regions, and so arguments of selective impact may cause confusion if the distinction between coding and noncoding is not stated.

5.

p3: "Also, proliferation assays performed in cancer cells demonstrated that the loss of most genes would have a detrimental fitness effect".

This statement may be seen as misleading, as very few somatic events entail the loss of both copies of a gene. Something that is not addressed in the Introduction is that the deleterious effect of most somatic mutations is already known to be buffered by the existence of two gene copies (see e.g. Martincorena et al. 2017). Together with the reduced relevance of most gene functions in somatic cells, this is possibly the most important explanation for the tolerance of high mutation burdens in cancer cells, and is of course central to the idea of wildtype allele amplification discussed in the paper, as this process operates by reinforcing such a 'buffer' against mutation. This should be discussed openly to avoid giving the impression that the authors' hypothesis that "somatic amplifications of wild-type copies of cancer-essential genes create a buffer against inactivation" is a completely novel concept, rather than an extension of the 'buffering' argument.

6.

In relation to the point above, it should be made clear that the three explanations which the authors present for the observation of high mutation burdens in tumours are neither independent nor mutually exclusive. The text gives the impression that the authors' gene amplification theory constitutes a novel explanation that competes against previous ideas like Muller's ratchet. On the contrary, by buffering the effect of highly deleterious mutations, and effectively rendering these mutations neutral or only slightly deleterious, gene amplification contributes to the phenomenon of Muller's ratchet, which refers to the irreversible accumulation of slightly deleterious mutations in clonal lineages. The amplification hypothesis can thus be seen as a mechanism for shifting the selective effects of mutations towards the range where Muller's ratchet is thought to operate. If there is a high rate of highly deleterious mutations, then negative selection will interfere with Muller's ratchet by directly purging mutant cells. See e.g. Gillespie's "Population Genetics: A Concise Guide".

7.

p4: "This analysis revealed a mutation-amplification-selection balance with intermediate-sized amplifications acting as optimal buffers".

The term "mutation-amplification-selection balance" seems to be used a few times as a sort of buzzword, but is not really introduced or defined at any point. The traditional mutation-selection balance refers to the situation where the number of deleterious alleles in a population is stable due to the mutation rate equalling the rate of mutation elimination by selection. How does the mutation-amplification-selection balance extend this idea? Is the number of deleterious alleles in a tumour constant due to amplification and selection? Is the number of amplified wildtype alleles constant due to mutation and selection? And is there actual evidence that this balance exists in cancers? Or perhaps it is not necessary to employ this term at all?

8.

p7: "The correlation differences were significant when comparing the correlation distributions in all considered cancer types (Fig. S3b ...)".

This sentence is inaccurate and should be corrected. Fig S3b does not show correlation differences per cancer type; it seems to show an aggregate for the 9 types considered here, and only the correlation difference between CADD (not PolyPhen) categories is significant.

9.

p9: "We observed that by summing up the three factors (OG + GO + μ score), we were able to increase the correlation with amplification frequency in both BRCA and PAAD (Fig. 3 and Fig. S4). As a control, we

swapped amplification frequencies between BRCA and PAAD, observing a drop in performance in both cases (Fig. S5), therefore confirming the tissue-specificity of the observed associations".

In addition to using Spearman correlations, I would recommend testing whether the addition of the μ score significantly improves the model by using a likelihood ratio test to compare the linear model (amplif freq \sim OG + GO + μ score) against the simpler model (amplif freq \sim OG + GO).

Furthermore, in order to verify that the tissue specificity of the association is partly derived from the μ score, it would be necessary to replicate the analysis shown in Fig S5 using the BRCA and PAAD μ scores on their own, rather than the sum (OG + GO + μ score).

10.

p10: "Even when pooling mutations within different karyotypes, our classification showed that overall half of mutations occur before the amplification events (50,2%); instead, when focusing on the mutations within single karyotypes, they appear after the amplifications in most cases (pie charts in Fig. 4 and Fig. S6)".

The use of the term "karyotype" in this context is not correct – a karyotype is obviously not a property of a genomic segment. What is called "karyotype" in Fig 4 could be rather called the segment's allele-specific copy number.

Also, the fact that half of the mutations in these tumours seem to predate the amplification events should be better addressed in the text, given the claim that "the majority of the karyotypes do not amplify the mutated segments".

11.

p11: "Conversely, protected gene sets are enriched in essential cellular functions relevant for tumor cell homeostasis, such as "organelle assembly and localization", "protein transport", "regulation of localization", "plasma membrane organization", "membrane phospholipid biosynthesis", "aerobic respiration", and "nucleotide metabolism" ($q < 0.05$; Fig. 5b)".

The fact should be discussed that GO terms related to DNA (DNA binding, DNA repair, replication, transcription), which correspond to very essential cellular processes, are absent from the set of protected genes (Table S3, sheet 3), but are instead found in the set of unprotected genes (sheet 6). This seems to contradict the authors' statement. It is also surprising that GO terms related to "cell cycle", "mitosis" and "apoptosis" are completely absent from both sets.

12.

p12: "Typically, the majority of somatic mutations in tumors has been considered to be fitness neutral. However, mounting evidence suggests that the majority of those mutations are associated with a fitness cost".

This sentence appears to be misleading. The fact that most somatic mutations are considered to be neutral is simply because they occur in non-coding regions. The claim that "the majority" of mutations are associated with a fitness cost only applies to mutations in coding (and maybe regulatory) regions, and this is not a controversial issue. It is thought that many passengers are slightly deleterious rather than selectively neutral, but their fitness effect is overcome by their linkage to driver mutations, which are under strong positive selection (a fact that does not seem to be discussed in the paper). Amplification events may be selectively advantageous insofar as they protect against *highly* deleterious mutations (e.g. mutations in haploinsufficient genes), rendering them neutral or slightly deleterious, and therefore tolerable when linked to positively selected drivers. This theoretical framework for the authors' findings – assuming they agree with it – should be made much more explicit in the paper.

13.

p12: "Our results suggest that somatic amplifications can act as a simple compensatory event to make tumor genomes more permissive for the accumulation of deleterious passenger mutations. This class of events constitutes a type of genomic aberration within cancer genome itself beyond the classical categorization in drivers and passengers".

I would disagree with this. Just like single-base substitutions or rearrangements, somatic amplifications can indeed be classified into driver and passenger events based on their role in enabling proliferation and adaptation. For instance, a somatic amplification is considered a driver if it amplifies an oncogene such as MYC, but is a passenger if it amplifies an intergenic region or entails a fitness cost. Therefore, it is not correct to say that "somatic amplifications" as a category are neither drivers nor passengers. In the specific case of an amplification that generates additional wildtype copies of a gene, it could be argued that some of these can be classified as drivers in some cancers, if it can be shown that they are significantly enriched in cancer genomes (i.e. that there is positive selection for these events). This is no different from the way in which cancer-driver genes have been identified by mutation enrichment analyses. From a statistical standpoint, it is likely that the vast majority of somatic copy number alterations are passenger events, just like the vast majority of single-base substitution events. The authors should consider correcting this statement.

14.

p13: "We hypothesized that larger amplification lengths (from 50 Mbp to chromosome level) have too high fitness cost compared to the deleterious impact of mutations; while smaller correlation lengths (from gene level to 20 Mbp) would require more try and error leading to a scattered somatic amplification landscape in order to cover most cancer-essential genes. In other words, amplifications will be selected whenever the fitness cost of deleterious mutations outweighs the cost of the amplifications, resulting in a mutation-amplification-selection balance (Fig. 6)".

This sentence seems unnecessarily problematic, as it implies (1) that all amplifications inevitably entail some fitness cost, and (2) that the size distribution of amplifications reflects only the action of selection, and not the cellular processes that give rise to these events. The authors should consider a more balanced interpretation.

15.

p18: "Genes were classified in protected and unprotected based on the protection index (P_i), calculated as: $P_i = \text{Amplification frequency} - \mu \text{ score}$ ".

What is the rationale for this formula? Are the amplification frequency and the $\mu \text{ score}$ defined on compatible scales?

16.

In relation to references 4 (Zapata et al.), 6 (Williams et al.) and 10 (Shen et al.), the authors should also consider the following relevant critiques of these studies:

Van den Eynden, J., Jiménez-Sánchez, A., Miller, M.L. et al. Lack of detectable neoantigen depletion signals in the untreated cancer genome. *Nat Genet* 51, 1741–1748 (2019).
<https://doi.org/10.1038/s41588-019-0532-6>

McDonald, T.O., Chakrabarti, S. & Michor, F. Currently available bulk sequencing data do not necessarily support a model of neutral tumor evolution. *Nat Genet* 50, 1620–1623 (2018).
<https://doi.org/10.1038/s41588-018-0217-6>

Balaparya, A., De, S. Revisiting signatures of neutral tumor evolution in the light of complexity of cancer genomic data. *Nat Genet* 50, 1626–1628 (2018). <https://doi.org/10.1038/s41588-018-0219-4>

Tarabichi, M., Martincorena, I., Gerstung, M. et al. Neutral tumor evolution?. *Nat Genet* 50, 1630–1633 (2018). <https://doi.org/10.1038/s41588-018-0258-x>

Kruglyak, L., Beyer, A., Bloom, J.S. et al. No evidence that synonymous mutations in yeast genes are mostly deleterious. *bioRxiv*. [bioRxiv 2022.07.14.500130](https://doi.org/10.1101/2022.07.14.500130) (2022).
<https://doi.org/10.1101/2022.07.14.500130>

Dhindsa, R.S., Wang, Q., Vitsios, D. et al. A minimal role for synonymous variation in human disease. *The American Journal of Human Genetics*, 109(12), 2105-2109 (2022).
<https://doi.org/10.1016/j.ajhg.2022.10.016>

Specifically, the authors should take into account the critique by Van den Eynden et al. when citing the findings of Zapata et al. (p3). The model used by Zapata et al. has been argued to produce spurious results due to its failure to account for the effects of sequence context in the mutation rate.

17.

I would encourage the authors to conduct simulation or permutation analyses in order to estimate the degree of positive selection for amplification of certain genes in the 9 cancer types highlighted in the paper. Although the correlations between mutation burden, amplification frequency and mutation score provide substantial evidence that this phenomenon is selectively advantageous in some tumours, it would be very interesting and useful to have quantitative estimates of the excess of observed amplification events around essential or cancer-relevant genes, relative to the expected background rate of amplification in these tumour types. Perhaps this could be achieved by using a simpler version of the kinds of ratios used to identify enrichment of certain mutation types relative to their neutral counterparts, such as the dN/dS ratio. Although this is only a suggestion, it would certainly enhance the relevance of the study to the cancer evolution field.

MINOR COMMENTS

18.

p2: "common measures of selection – such as the ratio of non-synonymous to synonymous substitutions (dN/dS)".

Note that dN/dS is actually the ratio of non-synonymous to synonymous substitution rates.

19.

p3: "This phenomenon has been already observed in different species such as the asexual amoebas and bacteria".

It would be more correct to say "Similar phenomena", as those in amoebae and bacteria are not exactly the same process as somatic copy number amplification.

20.

p3: "Moreover, whole-genome doubling – which preserves the ratio of mutant to wild-type alleles while doubling the number of wild-type ones – has been recently suggested to serve as a compensatory phenomenon for the accumulation of deleterious mutations in loss-of-heterozygosity regions".

Note that this argument applies to regions of "haploid LOH", and not so much to copy-neutral LOH (see López et al. 2020).

21.

p4: "considering only mutations within copy-neutral regions".

It would be useful to clarify in the Introduction that "copy-neutral" refers to regions present in two copies.

22.

p6: " $j = 36$ Mbp turned out to be the optimal segment length for further analysis since it displayed the best associations in terms of Spearman's correlation and associated p-value".

It could be clarified that " j " refers to the total length between both ends of the amplified segment (if that is the case). The use of the label "Gene" in the x-axis of Fig 1e makes it unclear whether this instead represents distance in either direction from the gene locus.

23.

p7: "Our relationship analysis demonstrated that frequently mutated haploinsufficient genes tend to be amplified during tumor evolution, which is likely to create a buffer to compensate for the loss of the gene product upon mutation".

Of note, this finding is consistent with Martincorena et al.'s (2017) results for cancer genomes, where they observed no detectable negative selection in haploinsufficient genes (Fig 3H), and negative

selection against mutations in essential genes located in haploid regions, but not in diploid regions (Fig 3G).

24.

p8: "Hence, we removed from the analysis the mutations within OGs and TSGs to assess whether they may inflate the correlations, which, in the end, were neither significantly decreasing nor increasing across all cancer types (Fig. 2d and Fig. S3a ...)".

This should be Fig. S3d, not S3a.

25.

p9: "Based on the analyses from Davoli et al. and Sack et al., we designed a modified version of the OG and GO scores ...".

A brief mention of what these scores measure and what they mean should be included in the main text. The Methods contain the formulas for these scores, but do not explain what they mean (or even the meaning of the abbreviations "GO" and "OG"). Note also that "GO" is being used with two different meanings in the paper, which might cause confusion.

26.

Table S3: Note that very small p-values are erroneously written with a positive exponent, e.g. "1.98E+10", rather than "1.98E-10".

Reviewer #5 (Remarks to the Author): Expert in cancer evolution, genomics and mathematical modelling

Alfieri et al presented a study linking genomic amplifications with accumulation of deleterious mutations across the cancer genomes from TCGA. The authors found that regions of the genome that tend to accumulate more deleterious mutations often have higher frequencies of amplifications. They found haplo-insufficient and essential genes tend to have better protection from amplifications. They also

explored this phenomenon in tissue specific manner. Subsequently, they dated the relative timing of amplifications and mutations. In the end, they performed functional enrichment analysis of the “essential” genes.

My biggest concern with this study is lacking of novelty. The relationship between genome instability/amplification and deleterious mutations have been known for many years, so, it is not a new hypothesis. The major novelty the authors developed is the correlation between local amplifications and mutations. Using this correlation metric, they explored potential connections to many different factors, each of which delivered findings with significant overlap with what we already know in the field and the statistical evidence are often not very strong and consistent across cancer types. In addition, the presentation is a collection of “positive results” from their explorations and the results are rather discrete. The link to existing literature is not as strong. I feel this might be a good paper for a professional journal. Here are my major concerns:

- 1) The “gap in the field” in the introduction is not very clear. As I mentioned earlier, the hypothesis the authors articulated is known for many years.
- 2) When the authors performed segment specific correlation analysis, was multiple test correction performed?
- 3) The haplo-insufficient genes, were discovered based on the same principle as the authors’ definition of compensatory effect. In some sense, the finding presented in Figure 2a is a bit circular.
- 4) The results surrounding Figure 2b (conservation score such as Polyphen and CADD) and Figure 2c (aggregating regions) are not consistent across cancer types. The discrepancies across cancer types significantly challenge the authors’ hypothesis and conclusion.
- 5) The section on tissue specific amplification pattern is very unclear and very hard to follow.
- 6) Timing of the mutational events (amplifications and mutations), the hypothesis isn’t as clear. As mutations are continuously accumulating (not a single event), the hypothesis the authors articulated is not as clear. In addition, the timing results for different karyotypes are very different in Figure 4. I am not sure the results are robust and consistent.
- 7) It is hard to sense the novelty from the functional enrichment of the protected gene list identified in the last section.

Reviewer #2

The manuscript of Alfieri et al. describes a correlation analysis between copy number amplification (CNA) and somatic mutation frequencies in cancer genomics data from The Cancer Genome Atlas (TCGA). Based on these correlations they suggest that CNAs are protective (or compensatory?) mechanisms for the deleterious effects of somatic mutations in (haploinsufficient) essential genes. This could explain why signals of negative selection have been found largely absent in cancer.

The manuscript is overall well-written and easy to follow. The starting hypothesis is very intriguing, and the authors made a substantial effort to provide several analyses in support of this hypothesis.

We thank the reviewer for the positive comments and interest in our work.

My main criticism is the purely correlation nature of the study. Obviously, correlations don't mean causation and in my experience are rarely what they seem in these kinds of genomic analyses. In this regard I was lacking some orthogonal (ideally wet lab) validation of the authors' findings. I was also surprised that the authors did not discuss the pure correlational nature as a limitation of their study.

My main concerns are the putative confounding effects of (1) genomic instability and/or (2) the trivial correlation between mutation rates and copy number variations (CNVs).

1.

The authors used one approach to exclude that genomic instability is a confounder, by focusing on copy number deletions (CNDs). While they found a (non-significant) similar correlation between mutations and deletions (fig. S2), when focusing on individual cancer types (fig. 1D) they found a clear difference between CNAs and CNDs. It seems though that the correlation becomes negative for deletions? Could the authors comment on this? Doesn't this simply indicate that less mutations are detected in (hemizygotously) deleted areas (only one chromosome segment) versus amplified areas (multiple segments). The authors tried to correct for this by only using mutations from copy number neutral segments, but as the classification as CNN/CNA/deletion depends on arbitrary cut-offs I'm concerned that this analysis indicates that the problem still remains.

Given these findings, I strongly disagree with the rather absolute statement that "these results discarded that possible confounding explanation ..." (line 120-121).

We thank the reviewer for raising this important point. As the reviewer acknowledges, we introduced the μ score in our analysis to minimize the impact of the obvious link between the number of genomic copies of a region and their likelihood of mutations occurring therein. We estimated the mutation rate (μ) from near-copy-neutral segments, but the exact definition of copy-neutrality is not straightforward as we need to satisfy a trade-off between either having very restrictive criteria and not accurately estimating the mutation rate or having looser criteria and then being influenced by the aforementioned bias. We agree with the reviewer that the observed negative correlations between mutation rate and CNA state could be indeed indicative of regions with fewer genomic copies having less chances to be mutated and hence questioning our model relying on the positive correlations.

In the previous version of the manuscript, we aimed to rule out the trivial association between copy numbers and mutation propensity by introducing a stringent cutoff (± 0.2), that is the `segment_mean`. This cutoff is more stringent than in several previous studies (e.g., in the TCGA copy-number detection pipeline is they use a cutoff of ± 0.3 (TCGA-CNV pipeline)).

In order to rule out that our observations could be explained by this trivial associations, we designed three controls:

(1) We asked which is the minimum cutoff interval for which we do not observe a significant relationship between mutation load and CNAs. To answer this question, we calculated the μ score using mutations within a symmetric `segment_mean` range and we correlated this μ score with the mean of `segment_mean` within those copy-neutral segments (as a proxy for amplification frequency). If the confounder were present for a specific `segment_mean` range, we should observe positive and significant correlations. Thus, we iteratively restricted the values of `segment_mean` to calculate the correlation between CN and number of mutations: e.g., $(-0.10/+0.10)$, $(-0.125/+0.125)$, $(-0.15/+0.15)$, $(-0.20/+0.20)$, ..., $(-0.85/+0.85)$, $(-0.90/+0.90)$. As shown in **Fig. R2.1a**, if we consider a statistical p-value cutoff of 0.05, the lowest non-significant correlation with higher `segment_mean` is the cutoff of ± 0.25 , which is more relaxed than what we used in the revised and the previous version of the manuscript. In addition, we showed that increasing the stringency by decreasing the `segment_mean` cutoff, strongly impacts the number of copy-neutral samples needed for the μ score calculation (**Fig. R2.1b**).

Figure R2.1

a: Correlations between the μ score and the mean of *segment_mean* within intervals spanning from \pm the given *segment_mean*. Below the correlation associated p-values, the dotted line represents the p-value significance cutoff of 0.05. **b:** Mean no. of patients available to calculate the μ score within a given *segment_mean* range (x-axis).

(2)
 We then compared the extent of correlations between the μ score and amplification frequency with the μ score and deletion frequency. As shown in **Fig. R2.2** and **Fig. R2.3**, the majority of cancer types show a higher extent of correlation with amplification frequency than with deletion frequency for different cutoffs (0.10, 0.15 and 0.20).

Figure R2.2

Ratios of correlations between amplification frequency and μ score over deletion frequency and μ score. Mutations were computed using different cutoffs of *segment_mean* (0.1, 0.15 and 0.2)

(3)

To further strengthen our results, we repeated the correlation analyses using two stringent cutoffs (0.10 and 0.15 - which also showed non-significant correlations as shown in **Fig. R2.1a**). As depicted in **Fig. R2.3-7** the results are very reproducible with both cutoffs, only with few exceptions, those latter are mainly present at the most stringent cutoff (0.1), which is likely because too few copy-neutral mutations were available within that interval (see **R2.1b**).

Figure R2.3 – Correlations between amplification or deletion frequency with the μ score
Spearman's correlation between amplification and μ score in all genes or deletion frequency and μ score in all genes. Mutations were computed using different cutoffs of *segment_mean* (0.1, 0.15 and 0.2)

Figure R2.4 – Correlations with the μ score calculated using haploinsufficient or non-haploinsufficient gene mutations
Spearman's correlation between amplification and μ score within haploinsufficient (LOF-intolerant) genes and μ score within non-haploinsufficient (LOF-tolerant) genes. Mutations were computed using different cutoffs of *segment_mean* (0.1, 0.15 and 0.2)

Figure R2.5 – Correlations with the μ score calculated using damaging or non-damaging mutations (CADD)
 Spearman's correlation between amplification and μ score using damaging or non-damaging mutations (using CADD) in all genes.
 Mutations were computed using different cutoffs of *segment_mean* (0.1, 0.15 and 0.2)

Figure R2.6 – Correlations with the μ score calculated using damaging or non-damaging mutations (PolyPhen)
 Spearman's correlation between amplification and μ score using damaging or non-damaging mutations (using PolyPhen) in all genes.
 Mutations were computed using different cutoffs of *segment_mean* (0.1, 0.15 and 0.2)

Figure R2.7 – Correlations with the μ score calculated using aggregation- and non-aggregation-causing mutations
 Spearman's correlation between amplification and μ score using aggregating or non-aggregating mutations in all genes.
 Mutations were computed using different cutoffs of *segment_mean* (0.1, 0.15 and 0.2).

To summarize, although we cannot entirely rule out the possibility of a confounding influence between copy number segments and mutation rate, our findings indicate that (1) this impact is minimal, and (2) it affects only some cancer types in a few comparisons, and the overall trend remains unaffected by this factor. We believe that our cutoff value (± 0.2) represents a good tradeoff between being restrictive enough to reduce the impact of genomic copy counts on mutation rate (**Fig. R2.1a**) and, on the other hand, having enough samples and mutations per segment (**Fig. R2.1b**).

In addition, we clarified in the revised version of the manuscript on lines 139-140 that the statement “these results discarded that possible confounding explanation” does not refer to the above-discussed potential impact of genomic amplifications on increased mutation rates by providing additional segments that could be independently mutated. Instead, it refers to our effort to eliminate a different confounder, namely a potential association between local mutation and double strand break rate.

2.

One additional negative control analysis that could be considered is a focus on mutations in non-expressed genes (like the other analyses shown in fig. 2). Correlations should be largely absent here if the authors' interpretation is correct. Additionally, the analyses shown in fig. 2 is now univariate only. How about interaction between these variables? E.g., non-damaging mutations in haploinsufficient genes?

We agree with the reviewer's expectation that selective pressure to buffer non-expressed genes with amplifications should be significantly reduced. To assess this, we conducted an analysis to calculate the μ score, utilizing independently mutations within expressed and non-expressed genes, while taking into account patient-averaged tumor type-specific transcriptomic data.

Indeed, we found that the reviewer's hypothesis was correct, and we observed a significant decrease in correlations (p-value = $5.8e-03$; paired one-sided Wilcoxon test) when analyzing only mutations in non-expressed genes, with the only exception being TCGA-LUSC. Considering the relevance of these findings, we have incorporated this analysis into the manuscript (lines 180-186 and in **Fig. 2c-d** and **Fig. S3c**), and it is now included in the main **Fig. 2c-d**.

In addition, as suggested by the reviewer, we combined the mutation impact predictor (CADD) within haploinsufficient and non-haploinsufficient genes. It is important to consider that, by subsetting genes and mutations therein, we dramatically reduced the number of observations in some conditions, making correlations prone to technical bias (e.g., sample size). However, as shown in **Fig. R2.8** we can observe an overall trend for non-damaging mutations within haploinsufficient genes to be less buffered than damaging mutations.

Moreover, consistent with this remark, please also see the reply to reviewer #3 first comment where we successfully subset synonymous and non-synonymous mutations within haploinsufficient and non-haploinsufficient genes.

Figure R2.8

Spearman's correlations between amplifications and damaging or non-damaging mutations (predicted using CADD) within haploinsufficient or non-haploinsufficient genes.

3.

Could the authors comment why the correlation was only significant in 9 cancer types (fig. 1C)? And why these exact cancer types? I find it remarkable that all 9 cancer types have a high mutational burden and/or TCGA sample size. Additionally, I would suggest that all downstream analyses (e.g., fig 1D, S3, ...) are performed on the complete set of cancer types and added as supplementary figures.

The reviewer raised an important point that was not sufficiently covered in the previous version of the manuscript. Additional explanatory sentences were added in the Discussion on lines 356-367.

“By analyzing genomic data from a large cohort of patients in TCGA and PCAWG, we found that amplifications can act as buffers that enable tumor genomes to tolerate the accumulation of deleterious passenger mutations. This buffering effect may help reduce the fitness cost of accumulating deleterious mutations, allowing the tumor to maintain growth and survival (Fig. 6). This effect depends on the tumor-specific mutation burden. In asexual populations as tumors, cells irreversibly accumulate mutations and could potentially experience a decrease in fitness as a consequence. In fact, due to their high mutation rate, the ratchet-like accumulation of deleterious mutations would increase the likelihood of population extinction through mutational meltdown. The higher the mutation rate, the faster the population may potentially become extinct, unless some buffering effect comes into play. As shown in Fig. 1f, taking correlations between the amplification frequency and the μ score as a proxy for buffering, we showed that this effect is increasing with higher mutation load.”

As the reviewer suggested, some factors could contribute to this effect, such as the amplification load (which is not statistically significant, but there is a positive trend as shown in **Fig. S2b**) or the TCGA sample size. To investigate the latter, and similar to what we have done in **Fig. S2b**, we tested the association between TCGA sample sizes and the correlations between the amplification frequency and the μ score, observing a non-significant but positive trend of 0.3046 with $p = 0.1576$ (Spearman's correlation test; not shown). While there is a stronger association with TMB, we conclude that biological factors explain better the which cancer types show significant correlations than technical factors.

As a request of the reviewer, we provided the analysis for the whole set of 23 TCGA cancer types as supplementary figure (**Fig. S3**).

4.

Based on their results, the authors interpret the amplifications as a protective mechanism for the deleterious effect of somatic mutations occurring later during tumor evolution (nicely summarized in fig. 6). However, on several occasions in the manuscript, they call the CNAs “compensatory events”. Isn’t this contradictory as this suggests CNAs compensate for mutations and hence occur later in tumor evolution? I found this very confusing. Additionally, assuming the CNAs occur first, could the authors speculate on the mechanism explaining the occurrence of these CNAs? Why would a cancer cell gain a CNA in the first place as it might cause a short-term fitness disadvantage as the authors indicate in fig 6?

We agree that some concepts were insufficiently explained and would need further clarifications. In addition, we do understand that the term “compensation” may lead to misinterpretation in the timeline of the events and decided to call it “buffering effect” instead. We use this term throughout the whole manuscript.

We would like to refer the reviewer to our reply of the 2nd comment of the 4th reviewer, who highlighted some ambiguous phrasing regarding this issue.

5.

Regarding the enrichment analysis shown in Fig. 5. I found it hard to interpret this figure and I don’t see the enrichments that are discussed in lines 254-259. The semantic spacing doesn’t improve interpretation here, so I would advise a simpler, more intuitive plot (e.g., barplots with ranked enrichment p values).

We agree with the reviewer that the enrichment analysis shown in **Fig. 5** was partly difficult to interpret. In response to this concern, we have adopted ReViGO treemap plots, as suggested, instead of ReViGO semantic space.

Moreover, we have improved the definition of protected and unprotected gene sets by implementing a more stringent analysis. Specifically, we used two symmetrical and stringent cutoffs (0.06 and 0.94) and computed the *Protection index (Pi)*-associated distribution percentiles through permutations (see Methods on lines 558-563). We then filtered the protected and unprotected gene sets using a distribution percentile cutoff below 0.1.

We invite the reviewer to refer to the updated **Fig. 5** and **Fig. S7**, as well as the revised Results section on lines 320-325 and 332-340 for further details.

6.

I found the methodology sometimes hard to understand, often lacking details necessary to understand how the analyses were performed exactly. Some examples: how is μ score calculated (explained in fig. S1 but more intuitive explanation needed in methodology), how are haplo(in)sufficient essential genes defined (+ how many are classified in each group), what is the OG/GO score (only possible to understand for a reader when reading the citations at the moment), ...

We understand the reviewer's concerns, and we accordingly decided to include **Supplementary Table ST4**, which contains the number of genes and mutations retrieved for each tumor type.

In the Method, we have added some sentences explaining how pLI is defined (lines 482-486). We have also expanded the description of the Methods section regarding the timing of somatic events (lines 545-552). Additionally, we specifically added some sentences to better explain how STOP and GO genes were defined (lines 230-235).

Some minor comments:

1. Fig. S1: color legend is lacking

We corrected this by adding the color legend in **Fig. S1**.

2. Fig. S2: “mutation score” in x axis. Is this the same as “ μ score”?

It is. We replace “mutation score” with “ μ score” in **Fig. S2a**.

3. Fig. 1e: “correlation length” should be “segmentation length”?

We replaced “Correlation length (j)” with “Segment length (j)” in **Fig. 1e**.

4. Many figures/analyses often lack “n=” details, e.g., how many haplo(in)sufficient, aggregating ... genes were found in fig. 2?

We refer the reviewer to Supplementary **Table ST4** added upon your previous comment.

5. Line 130: Where do I see p values in plot shown in fig. 1e. Only correlation coefficients are shown? Relatedly, several plots show (Pearson?) correlation coefficients, not just “correlations” as indicated in axes.

We added the p-value as a dotted line in **Fig. 1e**.

6. I found fig. S3 more useful than main fig. 2. The authors could consider swapping main and supplement here.

The reason why initially we showed only **Fig. 2** is that in the previous **Fig. S3** the identity of single tumor types is missing. However, we agree the overall trend is probably easier to understand from (previous) **Fig. S3** and, therefore, we merged **Fig. 2** and (previous) **Fig. S3** into the new **Fig. 2**.

7. The added value of the analysis shown in fig. 3 is limited. Why is this analysis only performed on 2 cancer types?

The limitation is that the GO score was experimentally computed with BRCA and PAAD cell lines only (through *in vitro* overexpression library). Further details were added in the main text to make this point clear on lines 230-235.

8. Added value of non-coding gene analysis in fig. 4 unclear. Which analysis does the Fisher test refer to? In general, I found the analysis in this figure hard to understand.

We understand the concern of the reviewer. We decided to move the histograms to the supplement (**Fig. S5**), and the pie charts were replaced with a more intuitive representation of contingency tables (**Fig. 4**).

9. Line 264: would add wilcox p value on plot

We added the p-value on the top of the boxplot as requested.

10. Line 435: above the 10th percentile. 90th percentile?

Thanks for the correction and yes, it was a mistake!

In conclusion, while I like the starting hypothesis of this extended analyses, additional evidence is required that demonstrates convincingly that the observed correlation is not biased, before this manuscript is publishable in Nature Communications.

Reviewer #3

Fabio Alfieri and colleagues suggested in the manuscript that copy number amplifications in cancer genomes could be used as a way to buffer damaging variants by creating multiple unmutated copies as backups. This idea has been proposed before, such as in Martincorena et al. Cell 2017, but here the authors provided several new lines of evidence to support this hypothesis: (1) mutation rate is positively correlated with amplification rate; (2) damaging mutations, haploinsufficient and essential genes tend to be more protected by amplifications; (3) amplifications tend to appear earlier than mutations during somatic evolution.

Overall, I found that this manuscript is easy to follow and quite interesting, and the majority of the results are supported by solid analysis. The topic of this paper would be found useful and important by readers from the cancer genomes community. However, I think there are a few major and minor points needing to be addressed to further improve the manuscript.

We thank the reviewer for the positive comments and interest in our work.

Major points:

1.

As the authors mentioned, the lack of negative selection was found by the dN/dS method. Have the authors tried to separate synonymous vs non-synonymous mutations when computing the association between the μ score and the amplification frequency in Fig. 1? Ideally, there should be stronger associations for non-synonymous mutations as synonymous mutations do not need the buffer. I would suggest the authors use synonymous mutations as their controls and compute $\mu_{\text{synonymous}}$ and $\mu_{\text{non-synonymous}}$ for their analysis. This was only partially answered in Fig. 2b. It's also related to Fig. 2a, I am wondering if the differential correlations for haploinsufficient genes when only predicted LOF mutations were used.

We agree with the reviewer's suggestion to compute the μ score separately for synonymous and non-synonymous mutations. As the reviewer correctly supposed, we observed weaker correlations when considering only synonymous mutations, except for TCGA-LUAD. Accordingly, this additional analysis of synonymous and non-synonymous mutations has been added in Fig.2 and Fig.S3 and a paragraph has been added in the Results section on lines 199-207.

Moreover, similar to our reply to the second comment of reviewer #2, we here separated synonymous and non-synonymous mutations within haploinsufficient and non-haploinsufficient genes. As shown in Fig. R3.1, we demonstrated that non-synonymous mutations have a higher buffering tendency compared to synonymous mutations within both haploinsufficient and non-haploinsufficient genes. Although the results appear strong, we need to highlight that in some categories there are few mutations, which make the results prone to noise. Therefore, we decided to not include it into the manuscript.

Figure R3.1

Spearman's correlations between amplification frequencies and synonymous or non-synonymous mutations computed within haploinsufficient or non-haploinsufficient genes separately.

2.

Similarly, the authors used PCAWG whole-genome sequencing data, so have the authors tried to compute mu scores for different categories of non-coding regions (promoters, enhancers, lncRNA and intergenic regions, etc.)? It would be interesting to see how the buffer effect changes among coding and non-coding elements.

We aimed to answer the reviewer comment using the PCAWG dataset and computed correlations with coding and non-coding mutations separately. We pooled projects by tumor type (<https://dcc.icgc.org/pcawg>) and removed tumor types with less than 100 patients with WGS data to ensure a reliable analysis. We calculated the μ score for coding and non-coding regions within diploid regions and correlated them with the amplification frequency. The results are presented as a main figure (**Fig. 4b** and **Fig. 4c**) and in lines 293-298 of the updated version of the manuscript. They show that non-coding mutations appear to be less strongly associated with amplification frequency than coding mutations.

3.

It has been shown in multiple previous studies that genomic features such as chromatin states, replication timing, GC content etc. have an impact on somatic mutation rate due to differential DNA repair and/or damage rates. If these features also affect amplification rates, then the correlation at the segment level can also be explained. Could the authors rule out this possibility?

The reviewer expressed concern regarding the potential common underlying causes of amplifications and mutation rates, which could explain the correlation between these two variables. This was also one of our initial concerns. To address this issue, we took two approaches:

a) We reasoned that if there were common features (*e.g.*, epigenomic factors) that affect the occurrence of amplifications and mutations, deletions should also be affected since they are both initiated by double-strand breaks. We therefore conducted a control experiment and tested for a correlation between point mutations and deletions. However, we found that they are largely independent.

b) We categorized mutations based on their impact (strong vs mild effect, haploinsufficiency, aggregation, etc.), as common covariates should equally affect those categories. Therefore, any differences we observed between these categories must be explained by other means.

4.

Many correlations observed by the authors were at the cohort or cancer type level (Fig. 1), but if copy number gains actually have the buffering function, they must co-occur with damaging mutations at the same genomic segment in the same individual. One would expect that, for example, the number of bases amplified should be positively correlated with the number of mutations across patients. I am wondering if the authors have observed this type of correlation as well.

Moreover, if the buffering hypothesis is true, direct speculation should be that at least some amplified segments should contain heterozygous damaging variants (synonymous mutations can be used as control) at the same individual. Could the authors find direct evidence for this and find out how widespread this phenomenon is?

We agree with the reviewer that copy number gains can have a buffering effect only if they occur before damaging mutations at the same genomic segment in the same individual. To test this, the reviewer proposed to correlate the number of amplified bases with the number of mutations across patients. Anyway, it is worth noting that this correlation is highly biased by the abovementioned confounder (we refer the reviewer to our reply to the first comment of reviewer #2): the higher probability of mutations to occur in amplified segments as the amplification will provide more copies that can be independently mutated.

We refer the reviewer to the plots below as an example (**Fig. R3.2**).

Figure R3.2
Correlations between amplification length and number of mutations across patients.

Despite this limitation, as suggested by the reviewer, we compared the synonymous and non-synonymous mutation burden across patients. As shown in **Fig. R3.3**, we observed that non-synonymous mutations tend to be more associated with amplification burden, supporting our buffering hypothesis.

Figure R3.3
Correlations between amplification length and number of all mutations, non-synonymous mutations and synonymous mutations across patients.

To answer the second part of the comment, we tried to understand how widespread this buffering effect is at tumor type and then at patient level., We would like to refer the reviewer to the updated **Fig. 4**, **Fig. S6d** and **Fig. S6e**, as well as to lines 293-309.

5.

The authors found an optimal segment length for amplifications, which is quite interesting. It looks like this optimal segment length is the optimal parameter for calculating the correlation, but not the optimal length of CNAs tumors can have. In fact, have the authors checked the distribution of CNA length (should be available from WGS data) and how frequent is the 36 Mbp length? Moreover, can the authors speculate the biological meaning of such an optimal length?

We thank the reviewer for raising this important point. Following their advice, we investigated the length distributions of CNAs across tumor types. Our analysis (**Fig. R3.4**) revealed that the majority of CNA lengths were below 36Mbp. One explanation for this difference is that there are several reasons why CNAs occur, and they may have varying effects on the length distribution. Our buffering model is just one of the possible factors that contribute to the observed length distribution. Other potential factors that could impact the length distribution are that CNA occurrence might be largely guided by random processes and many of them might not be selected but are merely tolerated by the genome. In addition, the density and distribution of OGs or TSGs contribute to their position and length. Overall, our findings suggest that the observed length distribution of CNAs in cancer types is likely influenced by a combination of different factors, including their role as buffers as well as other random and structural genomic properties. Further studies are needed to fully understand the complex mechanisms underlying CNA formation and their impact on cancer development and progression.

Figure R3.4

Distribution of CN amplifications (log10 of the length) within TCGA tumor types. The vertical black line represents 36Mbp.

6.

In lines 136-137, the authors implied that mutation accumulation could drive genome amplification. However, temporally, mutations should appear after the genome amplification given the buffering hypothesis. Could the authors elaborate a bit more on how biologically this can happen?

We agree that certain concepts were not properly explained and require further clarification. We understand that the terminology used may cause misunderstandings regarding the temporal order of events. Therefore, we have made several corrections in the manuscript to address this issue, such as on lines 20-21, 84-86, 101-104, 154-157 and 369-372. In particular, we need to clarify that we do not imply that mutation accumulation drives amplifications. Instead, our point is that cells which randomly acquire an amplification at a mutation-prone and cancer-essential location in the genome will have a growth advantage over cells that do not when mutations accumulate in this region.

We also refer the reviewer to the comment #2 of reviewer #4, who specifically highlighted some ambiguous phrasing regarding this issue.

7.

In Fig. 4, the analysis of mutation-amplification relative timing, it looks like the difference between coding and non-coding mutations is small. As in point 1, what if the authors separate synonymous and nonsynonymous mutations? Would there be a larger difference?

We thank the reviewer for this interesting idea. To address this, we compared the relative distributions of late and early mutations within synonymous and non-synonymous mutations. We added this new analysis in the Results section (lines 293-296) and show it in **Fig. S6b**.

Minor points:

1. In line 99, the authors mentioned that “the obvious confounder that the more copies of a segment exist the more mutations the segment could acquire”. I am wondering if the authors have also calculated mu scores for amplified or deleted regions.

In order to validate our initial intuition that copy numbers directly affect mutation rates as genomic segments should mutate independently from each other, we computed the μ score including mutations within both deleted and amplified segments.

As expected, we observed a strong absolute increase in correlations while associating the amplification or deletion frequency with the μ score calculated using mutations in amplified and deleted segments (Fig. R3.5). This suggests that the copy number changes directly influence the mutation rates and justify the use of our μ score as a valid method to infer segment mutations rate that largely reduces the impact of this confounder.

We would like to draw the reviewer’s attention to the first comment of reviewer #2, where we extensively discussed the threshold used in the analysis and how this could create biases in the correlation analysis.

Figure R3.5

Correlations between the μ score calculated using copy-neutral or amplified and deleted mutations with amplification or deletion frequencies.

2. The mu score measurement of mutation frequency is crucial to many conclusions in the paper. In many places, such as Fig.1b, the authors used the mean mu score. Since in many tumor types, there are hypermutated individuals (mutation rate >30/Mb), the mean mu score may be skewed by them. I am wondering if the authors have filtered out hypermutated samples.

We thank the reviewer for pointing out this potential confounder, which we addressed by conducting the analysis presented in Figure Fig. R3.6 and R3.7. Specifically, we excluded hypermutated patients (using the definition of the reviewer as those patients having more than 30 mutations/Mb, which removed up to 20 samples per tumor type) and we compared the μ score removing hypermutated samples and the μ score considering all samples. As shown in Fig. 3.6 where we presented the comparison for nine cancer types, we can observe overall strong correlations indicating that the removal of hypermutated samples does not profoundly affect our μ score calculation.

Figure R3.6
Correlations between the μ score using all patients and the μ score calculated by removing hypermutated samples.

Moreover, we replicated the analyses presented in Fig. 1b and Fig. 1c in Fig. R3.6a and Fig. 3.7b, respectively. Our findings demonstrated that correlations remain robust, indicating that including hypermutated samples has a minimal impact on the results.

Figure R3.7
a: replica of Fig. 1b removing hypermutated samples.
b: replica of Fig. 1c comparing single tumor type correlations using Fig. 1c (as control) and the ones removing hypermutated samples.

3. In Fig.1d, it looks like deletions have negative correlations with the mu score at the segment level. Do the authors have an explanation for this?

We would like to refer the reviewer to our reply to the first comment of reviewer #2.

4. In the section below line 247, could the authors provide the number of protected and unprotected genes, common essential and nonessential genes, as well as their overlaps?

We refer the reviewer to the new **Table ST3**, where we provided the full list of common essential, non-essential, protected and unprotected gene sets, as well as their overlaps.

5. In line 255, the authors mentioned “most strongly enriched terms were “olfactory receptors”, “sensory perception of taste” and “heart cell development and morphogenesis””, but not all terms are obvious in Fig. 5a.

We decided to change the visualization of the enrichment analysis using treemap plots (from rrvgo package in R), as well as the design of analysis itself. Consequently, some terms may have changed.

We refer the reviewer to the **Table ST3** and the updated results of the enrichment analysis shown at lines 320-326, **Fig. 5** and **Fig. S7**.

6. In Methods, the authors stated that “the genes above the 10th upper percentile of Pi were classified as protected, and below the 5th percentile as unprotected”. Why use these asymmetrical cut-offs (10th and 5th)?

We updated the way we define protected and unprotected genes in several ways. This includes that we have now symmetric cutoffs. We would like to refer the reviewer to the comment #5 of reviewer #2 for a description of the way we changed the analysis (and in addition in the manuscript lines 558-563).

Reviewer #4

Alfieri et al. present an interesting study of the role of somatic copy number amplification in cancer, particularly in terms of its contribution to the protection of functional genes from mutation. While the question of whether gene duplication is beneficial against mutation is certainly not new (e.g. PMID: 7768451, 17954278, 32139907), these authors provide unprecedented genomic evidence in favour of the hypothesis that amplification can be advantageous to cancer cells not only by amplifying oncogenes, but also by generating additional intact copies of important genes.

The results of this study are of relevance to the cancer research field, especially as they highlight one mechanism by which tumour cells may become tolerant of the effects of their own genomic instability – although such a mechanism is not as radically novel as the authors seem to suggest. The methodology of the study also seems adequate in general. I am more concerned, however, by the article's notable lack of clarity and preciseness with respect to a number of its central points. As it is, the manuscript seems rather ambiguous in expressing the authors' interpretations of their findings. Specifically, it is often unclear whether the authors are defending a surprising sort of teleology, whereby gene amplification is somewhat purposefully produced with the aim of protecting important genes against future mutations. It is really only near the end of the article that one becomes reasonably sure of understanding the authors' argument – which is fortunately not the one I have just outlined. Moreover, some of the most central concepts of the paper (the " μ score" and "mutation-amplification-selection balance") are ill-defined or not defined at all, with the authors seemingly taking their meaning for granted. Therefore, I think it is especially important that these issues of intelligibility be addressed by the authors, to avoid the article from conveying misleadingly inaccurate arguments.

Below I list a series of major and minor points which I believe should be addressed before publication. In most cases, I cite the relevant sentence in the manuscript before my comments.

We thank the reviewer for the positive comments and interest in our work.

Major comments:

1.

The concept of the " μ score", which is central to the paper and is mentioned repeatedly since the Introduction, is surprisingly not defined anywhere in the main text. Moreover, its alternative name of "mutation score" is only used in the legend to Figure 1 and the Methods, compounding the confusion. It is even difficult to find the definition and formula for this score in the Methods. Such a lack of detail about one of the most important concepts in the paper is quite concerning. The authors should address this problem by taking care to:

- Introduce the concept of the μ score at a suitable early point in the main text.
- Make it clear in the Methods that Eq. 2 refers to the μ score.
- Elaborate their rationale for the μ score formula. Is this an original formula? If so, why was it necessary to develop it, and what is the strength of this particular measure? If not, what is the original source of this score?

We thank the reviewer for bringing our attention to our incoherent usage of those terms. We consistently replaced “mutation score” with “ μ score” (Fig. 1 and lines 588 [within the legend of Fig. 1], 611 [caption of Fig. 3], and 655 [within the legend of Fig. S1]).

In the revised version of the manuscript, we introduced the μ score concept in the Introduction explaining the rationale behind this formula on lines 90-94 in an expanded manner. We make it explicit that Eq. 2 refers to the μ score by modifying Fig. S1 and line 465. We refer the reviewer to the first comment of reviewer #2 where we extensively addressed this point.

2.

I have encountered several instances where the authors' phrasing is notably ambiguous, as it suggests that they might believe copy number amplifications events to be explicitly "aimed" at protecting these regions from mutation, or even that they are triggered by mutation itself. After reading the entire paper, it is clearer to me that the authors believe in the (likely correct) view that these amplifications are random events which precede the mutations they are thought to protect against, and that it is due to selection during tumour evolution that these amplifications end up being overrepresented around certain types of genes. However, while one is in the process of reading the paper, the theoretical framework of the authors is not at all clear, and nearly every discussion of their results can easily be construed as a defense of the view that cancer genomes are deploying these amplifications in a strategic fashion in response to mutation. I would therefore encourage the authors to rework the text very carefully in order to avoid these unintended ambiguities. Below I list some such cases of ambiguous phrasing.

We thank the reviewer for their interest and commitment shown in these comments. We completely agree that very often the explanation or discussion of our hypothesis was lacking the contextualization of our model in a clear evolutionary framework. We corrected the manuscript accordingly as discussed in response to the following points:

- p1: "undergo frequent somatic amplifications of their wild-type form to protect them from deleterious mutations". This phrasing implies that protection of these genes is the cause of the amplification, rather than its consequence.

We modified lines 20-21 as follows:

“Ultimately, our work paves the way for the detection of novel cancer vulnerabilities by revealing genes that fall within amplifications likely selected during evolution to mitigate the effect of mutations.”

- p4: "we demonstrated that cancer-relevant regions undergoing high rates of damaging mutations tend to be amplified". Is this phrasing accurate? Presumably tumour-suppressor genes (which are also relevant to cancer) do not tend to be amplified. More importantly, the sentence may be read as arguing that these regions are duplicated more frequently (even in the absence of selection), rather than their amplification being selected more frequently.

We modified lines 101-104 as follows:

“Therefore, we demonstrated that regions highly susceptible to deleterious mutations and containing cancer-essential genes are often amplified during tumor evolution. This indicates that those amplifications might give a selective advantage to tumor cells to protect their genome against the detrimental effects of mutations and to maintain their fitness.”

- p4: "the hypothesis that cancer evolution selects somatic amplification events to mitigate the irreversible accumulation of deleterious mutations by creating extra safe copies of highly mutated regions".

I think this would be better phrased as "somatic amplification events are selected in cancer evolution, as they mitigate...". The sentence as it is seems to suggest that cancer evolution intentionally produces these events beforehand to mitigate the future accumulation of mutations. I also think that "intact copies of genes found in highly mutated regions" would be more correct than "safe copies of highly mutated regions", as the term "safe" is not ill-defined in this context.

We modified lines 84-86 as follows:

“Given all the previous observations, we sought out to investigate the hypothesis that somatic amplifications are selected in cancer evolution as they can buffer the subsequent gene inactivation through loss-of-function (LOF) mutations in cancer-relevant regions of the genome.”

- p6: "this suggests that amplifications may be driven by mutation accumulation and not vice versa". This may be read as implying a causal relationship ("driven") between mutation and amplification, which would of course contradict the finding that amplifications tend to precede mutations in those regions.

We modified lines 154-157 as follows:

“In contrast, having a higher amplification burden does not necessarily indicate a strong correlation, as depicted in Fig. S2b. This suggests that amplifications are likely to have been selected during tumor evolution in patients with high mutation rates rather than mutation rates being elevated as a consequence of more genomic copies.”

- p12: "A lot of species selects CN increase as a rescue mechanism to escape from the ratchet-like accumulation of deleterious mutations: horizontal gene transfer in bacteria, polyploidy in amoebas and whole genome doubling in cancer, which are all mechanisms aiming to the creation of safe wild-type copies of genes".

Besides needing some polishing, this sentence should be rephrased to avoid the message that amplifications have the "aim" of creating back-up copies of genes.

We modified lines 379--382 as follows:

“Indeed, many species adopt similar mechanisms by increasing genomic copy numbers to counteract the ratchet-like accumulation of deleterious mutations. Some examples of these mechanisms include horizontal gene transfer in bacteria and polyploidy in amoebas or yeasts. The same role has been hypothesized for whole genome doubling in cancer.”

3.

p9: "From a tumor evolution perspective, amplifications could exert a compensatory effect, buffering the deleterious consequences of mutations, only if they temporally appear before somatic mutations. Thus, only if CNAs amplify wild-type regions, they can render the segments more tolerant to mutation accumulation".

In relation to the previous point, it would be very helpful to introduce this idea in the Introduction. As explained above, it is currently not clear (until this point in the text) whether the hypothesis being proposed is that amplifications occur in wildtype regions and are later selected for their buffering effect, or alternatively that amplifications occur specifically in response to mutation.

We refer the reviewer to our reply to their comment #6.

4.

p2: "However, in many systems – including cancer cells – the majority of mutations have been observed to have a deleterious role on cellular fitness ... Specifically, among the different mutation types, more than 75% are non-synonymous ...".

It should be made very clear that this refers to mutations in coding-protein regions. This is an important point when speaking of the neutrality or deleteriousness of passenger mutations, as the vast majority of passenger mutations genome-wide are in noncoding regions, and so arguments of selective impact may cause confusion if the distinction between coding and noncoding is not stated.

We now specifically state in the revised version of the manuscript that we are referring to coding mutations only (lines 28 and 32).

5.

p3: "Also, proliferation assays performed in cancer cells demonstrated that the loss of most genes would have a detrimental fitness effect". This statement may be seen as misleading, as very few somatic events entail the loss of both copies of a gene. Something that is not addressed in the Introduction is that the deleterious effect of most somatic mutations is already known to be buffered by the existence of two gene copies (see e.g. Martincorena et al. 2017). Together with the reduced relevance of most gene functions in somatic cells, this is possibly the most important explanation for the tolerance of high mutation burdens in cancer cells, and is of course central to the idea of wildtype allele amplification discussed in the paper, as this process operates by reinforcing such a 'buffer' against mutation. This should be discussed openly to avoid giving the impression that the authors' hypothesis that "somatic amplifications of wild-type copies of cancer-essential genes create a buffer against inactivation" is a completely novel concept, rather than an extension of the 'buffering' argument.

We agree with the reviewer that the idea that ploidy affects selection is not new. However, we believe that relating the observed amplification patterns of a tumor (beyond diploidy) to cancer type-specific mutation rates and cancer gene-essentiality is. We mention now more clearly in the Introduction that diploidy has been discussed to relax negative selection (lines 60-63). Also, we now expand the discussion of the previous literature and the novelty of our ideas substantially (lines 349-410). We exemplify this in reply to the next comment.

6.

In relation to the point above, it should be made clear that the three explanations which the authors present for the observation of high mutation burdens in tumours are neither independent nor mutually exclusive. The text gives the impression that the authors' gene amplification theory constitutes a novel explanation that competes against previous ideas like Muller's ratchet. On the contrary, by buffering the effect of highly deleterious mutations, and effectively rendering these mutations neutral or only slightly deleterious, gene amplification contributes to the phenomenon of Muller's ratchet, which refers to the irreversible accumulation of slightly deleterious mutations in clonal lineages. The amplification hypothesis can thus be seen as a mechanism for shifting the selective effects of mutations towards the range where Muller's ratchet is thought to operate. If there is a high rate of highly deleterious mutations, then negative selection will interfere with Muller's ratchet by directly purging mutant cells. See e.g. Gillespie's "Population Genetics: A Concise Guide".

We thank the reviewer for the suggestions, we modified the Introduction on lines 54-71 as follows:

“The second explanation for the lack of detectable negative selection – though not independent from the first one – could be a consequence of the asexual nature of cancer evolution: in the absence of recombination, populations irreversibly accumulate mutations, and cells rapidly experience a fitness decrease - a process called Muller’s ratchet²⁴. In populations of finite size, with large genomes and high mutation rates, the ratchet-like accumulation of deleterious mutations increases the likelihood of population extinction through mutational meltdown²⁵⁻²⁷. This again suggests that tumors may evolve strategies to buffer for deleterious mutations. Diploidy, that is, the presence of two copies of each gene, is certainly the most important evolutionary invention providing

protection against mutation-induced gene inactivation^{28,29}. However, in specific cases where the overall mutation load is higher – such as in cancer cells – diploidy might not be enough to buffer deleterious mutations³⁰.

Nonetheless, we here propose a third complementary answer: cancer somatic copy-number amplifications can reinforce the tolerance towards deleterious mutations, countering the phenomenon of Muller’s ratchet. During cancer evolution, CNAs are randomly acquired in the genome, and when they temporally occur before somatic mutations, they create intact copies of the genes therein. Those amplifications can protect against the subsequent deleterious and irreversible accumulation of mutations, and then they would be selected and over-represented in mutation-prone regions with cancer-relevant genes.”

7.

p4: "This analysis revealed a mutation-amplification-selection balance with intermediate-sized amplifications acting as optimal buffers". The term "mutation-amplification-selection balance" seems to be used a few times as a sort of buzzword, but is not really introduced or defined at any point. The traditional mutation-selection balance refers to the situation where the number of deleterious alleles in a population is stable due to the mutation rate equalling the rate of mutation elimination by selection. How does the mutation-amplification-selection balance extend this idea? Is the number of deleterious alleles in a tumour constant due to amplification and selection? Is the number of amplified wildtype alleles constant due to mutation and selection? And is there actual evidence that this balance exists in cancers? Or perhaps it is not necessary to employ this term at all?

We thank the reviewer for the comment and appreciate their feedback. We revised the manuscript accordingly by removing the term “mutation-amplification-selection balance”. We understand that using it may require further evidence and may have caused confusion. Therefore, we agree that it is not helpful to employ this term in our study and removed it thorough the revised version of the manuscript.

8.

p7: "The correlation differences were significant when comparing the correlation distributions in all considered cancer types (Fig. S3b ...)". This sentence is inaccurate and should be corrected. Fig S3b does not show correlation differences per cancer type; it seems to show an aggregate for the 9 types considered here, and only the correlation difference between CADD (not PolyPhen) categories is significant.

We corrected this inaccuracy in the main text on lines 194-197.

9.

p9: "We observed that by summing up the three factors (OG + GO + μ score), we were able to increase the correlation with amplification frequency in both BRCA and PAAD (Fig. 3 and Fig. S4). As a control, we swapped amplification frequencies between BRCA and PAAD, observing a drop in performance in both cases (Fig. S5), therefore confirming the tissue-specificity of the observed associations".

In addition to using Spearman correlations, I would recommend testing whether the addition of the μ score significantly improves the model by using a likelihood ratio test to compare the linear model (amplif freq \sim OG + GO + μ score) against the simpler model (amplif freq \sim OG + GO).

Furthermore, in order to verify that the tissue specificity of the association is partly derived from the μ score, it would be necessary to replicate the analysis shown in Fig S5 using the BRCA and PAAD μ scores on their own, rather than the sum (OG + GO + μ score)

We thank the reviewer for raising this important question. It is worth noting that all the correlation analysis in the manuscript were performed using Spearman’s rank correlation coefficient. This choice was made as we observed mostly non-linear relationships between μ score and amplification frequency. Two examples are given in **Fig. R4.1**.

Figure R4.1

Examples of non-linear but monotonic correlations between the μ score (y-axis) and amplification frequency (x-axis) in BRCA and PAAD ($j = 36\text{Mbp}$).

Anyway, we understand that linear models are a powerful tool to simultaneously test the association between several variables. For this reason, we tested what reviewer proposed by creating the following linear models:

1. *linear model (amplification frequency ~ μ score)*
2. *linear model (amplification frequency ~ OG score + GO score)*
3. *linear model (amplification frequency ~ OG score + GO score + μ score)*

The results of the model performance (#1-3) are shown in **Fig. R4.2a** where we observed an increase for model #3, though not significant.

We reasoned that the lack of significance may be due to the assumption of linearity done by using the linear models. In order to remove the abovementioned limitation and only testing the monotonicity of our data, we fit the following linear models by rank-transforming the data, as:

4. *linear model (rank(amplification frequency) ~ rank(μ score))*
5. *linear model (rank(amplification frequency) ~ rank(OG score) + rank(GO score))*
6. *linear model (rank(amplification frequency) ~ rank(OG score) + rank(GO score) + rank(μ score))*

As shown in **Fig. R4.2b**, in both the multiple linear and ranked models, we observed a strong increase in performance.

Figure R4.2

Correlations using linear models or ranked linear models.

Moreover, as suggested by the reviewer, we performed the likelihood ratio test between models #2 and #3, obtaining a non-significant increase ($p = 0.1132$). However, we observed a significant improvement in the comparison of models #4 and #5 when we added the μ score ($p = 5.765e-3$).

Furthermore, as requested by the reviewer, we created a control by swapping the μ score between PAAD and BRCA (rather than only swapping the sum OG+GO+ μ score) and we modified lines 251-253 accordingly. We added these controls in the main **Fig. 3**. Furthermore, we changed the visualization of models in **Fig. 3** to be consistent with the vertical histograms shown in **Fig. 2**.

10.

p10: "Even when pooling mutations within different karyotypes, our classification showed that overall half of mutations occur before the amplification events (50,2%); instead, when focusing on the mutations within single karyotypes, they appear after the amplifications in most cases (pie charts in Fig. 4 and Fig. S6)".

The use of the term "karyotype" in this context is not correct – a karyotype is obviously not a property of a genomic segment. What is called "karyotype" in Fig 4 could be rather called the segment's allele-specific copy number.

Also, the fact that half of the mutations in these tumours seem to predate the amplification events should be better addressed in the text, given the claim that "the majority of the karyotypes do not amplify the mutated segments".

We agree with the reviewer that our use of the term “karyotype” is not correct in this context. Accordingly, we have revised this incorrect terminology throughout the manuscript using the reviewer's suggestion (“segment’s allele-specific copy number”).

Regarding the second part of the comment about mutation timing, we have addressed this issue in the manuscript by adding some new analyses that assess the degree of buffering within tumor types and patients. We refer the reviewer to the updated **Fig. 4** and new **Fig. S6**, as well as to lines 293-309.

11.

p11: "Conversely, protected gene sets are enriched in essential cellular functions relevant for tumor cell homeostasis, such as “organelle assembly and localization”, “protein transport”, “regulation of localization”, “plasma membrane organization”, “membrane phospholipid biosynthesis”, “aerobic respiration”, and “nucleotide metabolism” ($q < 0.05$; Fig. 5b)".

The fact should be discussed that GO terms related to DNA (DNA binding, DNA repair, replication, transcription), which correspond to very essential cellular processes, are absent from the set of protected genes (Table S3, sheet 3), but are instead found in the set of unprotected genes (sheet 6). This seems to contradict the authors' statement. It is also surprising that GO terms related to "cell cycle", "mitosis" and "apoptosis" are completely absent from both sets.

We appreciate the reviewer’s comment. We first of all would like to highlight that we changed the way that protected and unprotected genes were identified and functional enrichment among the calculated. We would like to refer them to comment #5 of reviewer #2 for motivating those changes better and giving details of the modifications we did to the method and its description in the manuscript. Those changes slightly change the enrichment in, as we believe, way that produces more stringently functional associations as expected from cancer-related and unrelated gene sets.

However, still some associations might be still surprising, and we asked how, for example, comes that there are transcription factors (TFs) among the unprotected genes. We observed that many of the unprotected TFs were in fact highly tissue-specifically expressed while protected TFs were not (**Fig. R4.3**). We therefore tested if in general protected genes were expressed in more tissues and in average at higher levels using both expression data from TCGA (**Fig. S9**) and GTEx (**Fig. R4.4**). We indeed observed that genes that are only expressed in few tissues or at low abundance and hence are more likely dispensable by the tumor, were less likely protected by amplifications from mutations (**Fig. 5d**).

In addition, we kindly refer the reviewer to new analyses described on lines 335-340.

Figure R4.3
Distribution across tumor types of highly expressed TFs within the unprotected gene set.

Figure R4.4
Distribution of gene expression values (using GTEx) among the protected and unprotected gene sets; and between the TFs within the protected and unprotected gene sets.

12.

p12: "Typically, the majority of somatic mutations in tumors has been considered to be fitness neutral. However, mounting evidence suggests that the majority of those mutations are associated with a fitness cost".

This sentence appears to be misleading. The fact that most somatic mutations are considered to be neutral is simply because they occur in non-coding regions. The claim that "the majority" of mutations are associated with a fitness cost only applies to mutations in coding (and maybe regulatory) regions, and this is not a controversial issue. It is thought that many passengers are slightly deleterious rather than selectively neutral, but their fitness effect is overcome by their linkage to driver mutations, which are under strong positive selection (a fact that does not seem to be discussed in the paper). Amplification events may be selectively advantageous insofar as they protect against *highly* deleterious mutations (e.g. mutations in haploinsufficient genes), rendering them neutral or slightly deleterious, and therefore tolerable when linked to positively selected drivers. This theoretical framework for the authors' findings – assuming they agree with it – should be made much more explicit in the paper.

We revised the manuscript to make the point clearer on lines 349-354.

13.

p12: "Our results suggest that somatic amplifications can act as a simple compensatory event to make tumor genomes more permissive for the accumulation of deleterious passenger mutations. This class of events constitutes a type of genomic aberration within cancer genome itself beyond the classical categorization in drivers and passengers".

I would disagree with this. Just like single-base substitutions or rearrangements, somatic amplifications can indeed be classified into driver and passenger events based on their role in enabling proliferation and adaptation. For instance, a somatic amplification is considered a driver if it amplifies an oncogene such as MYC, but is a passenger if it amplifies an intergenic region or entails a fitness cost. Therefore, it is not correct to say that "somatic amplifications" as a category are neither drivers nor passengers. In the specific case of an amplification that generates additional wildtype copies of a gene, it could be argued that some of these can be classified as drivers in some cancers, if it can be shown that they are significantly enriched in cancer genomes (i.e. that there is positive selection for these events). This is no different from the way in which cancer-driver genes have been identified by mutation enrichment analyses. From a statistical standpoint, it is likely that the vast majority of somatic copy number alterations are passenger events, just like the vast majority of single-base substitution events. The authors should consider correcting this statement.

We apologize for any confusion caused. We agree that our previous wording may have been misleading and could have been interpreted as suggesting that all amplifications are compensatory, which would negate the driver/passenger distinction for all types of somatic alterations.

We now state that the buffering potential of amplifications is another way of categorizing the type of effect of an alteration, in addition to classified as a driver or passenger. We have removed the statement that could have been misleadingly read as that all amplifications fall under this category. We refer the reviewer to the Discussion section on lines 369-372.

14.

p13: "We hypothesized that larger amplification lengths (from 50 Mbp to chromosome level) have too high fitness cost compared to the deleterious impact of mutations; while smaller correlation lengths (from gene level to 20 Mbp) would require more try and error leading to a scattered somatic amplification landscape in order to cover most cancer-essential genes. In other words, amplifications will be selected whenever the fitness cost of deleterious mutations outweighs the cost of the amplifications, resulting in a mutation-amplification-selection balance (Fig. 6)".

This sentence seems unnecessarily problematic, as it implies (1) that all amplifications inevitably entail some fitness cost, and (2) that the size distribution of amplifications reflects only the action of selection, and not the cellular processes that give rise to these events. The authors should consider a more balanced interpretation.

We understand the importance of the concern of the reviewer. We refer the reviewer to our reply to comment #5 of reviewer #3.

15.

p18: "Genes were classified in protected and unprotected based on the protection index (Pi), calculated as: $Pi = \text{Amplification frequency} - \mu \text{ score}$ ".

What is the rationale for this formula? Are the amplification frequency and the $\mu \text{ score}$ defined on compatible scales?

The rationale for the formula used to calculate the protection index (Pi) is to determine whether a gene is protected or unprotected based on its amplification frequency relative to the mean μ score. Both the amplification frequency and the μ score were defined on compatible scales, as they are both calculated using the same genomic region, i.e., the gene length. By defining both quantities on the same scale, we can compare the amplification frequency of each gene to the mean μ score and determine whether it is above or below the threshold value (94th or 6th quantile for protected and unprotected, respectively). This allows us to identify which genes are protected, meaning they have a higher amplification frequency than the μ score and therefore are more likely to be essential, and which genes are unprotected, meaning they have a lower amplification frequency than mean μ score and are less likely to be essential.

16.

In relation to references 4 (Zapata et al.), 6 (Williams et al.) and 10 (Shen et al.), the authors should also consider the following relevant critiques of these studies:

Van den Eynden, J., Jiménez-Sánchez, A., Miller, M.L. et al. Lack of detectable neoantigen depletion signals in the untreated cancer genome. *Nat Genet* 51, 1741–1748 (2019). <https://doi.org/10.1038/s41588-019-0532-6>

McDonald, T.O., Chakrabarti, S. & Michor, F. Currently available bulk sequencing data do not necessarily support a model of neutral tumor evolution. *Nat Genet* 50, 1620–1623 (2018). <https://doi.org/10.1038/s41588-018-0217-6>

Balaparya, A., De, S. Revisiting signatures of neutral tumor evolution in the light of complexity of cancer genomic data. *Nat Genet* 50, 1626–1628 (2018). <https://doi.org/10.1038/s41588-018-0219-4>

Tarabichi, M., Martincorena, I., Gerstung, M. et al. Neutral tumor evolution? *Nat Genet* 50, 1630–1633 (2018). <https://doi.org/10.1038/s41588-018-0258-x>

Kruglyak, L., Beyer, A., Bloom, J.S. et al. No evidence that synonymous mutations in yeast genes are mostly deleterious. *bioRxiv*. [bioRxiv. bioRxiv 2022.07.14.500130](https://doi.org/10.1101/2022.07.14.500130) (2022). <https://doi.org/10.1101/2022.07.14.500130>

Dhindsa, R.S., Wang, Q., Vitsios, D. et al. A minimal role for synonymous variation in human disease. *The American Journal of Human Genetics*, 109(12), 2105-2109 (2022). <https://doi.org/10.1016/j.ajhg.2022.10.016>

Specifically, the authors should take into account the critique by Van den Eynden et al. when citing the findings of Zapata et al. (p3). The model used by Zapata et al. has been argued to produce spurious results due to its failure to account for the effects of sequence context in the mutation rate.

We thank the reviewer for their precious suggestions. We added a sentence to the Introduction (line 36) highlighting the controversy about negative selection and referencing the relevant literature.

17.

I would encourage the authors to conduct simulation or permutation analyses in order to estimate the degree of positive selection for amplification of certain genes in the 9 cancer types highlighted in the paper. Although the correlations between mutation burden, amplification frequency and mutation score provide substantial evidence that this phenomenon is selectively advantageous in some tumours, it would be very interesting and useful to have quantitative estimates of the excess of observed amplification events around essential or cancer-relevant genes, relative to the expected background rate of amplification in these tumour types. Perhaps this could be achieved by using a simpler version of the kinds of ratios used to identify enrichment of certain mutation types relative to their neutral counterparts, such as the dN/dS ratio. Although this is only a suggestion, it would certainly enhance the relevance of the study to the cancer evolution field.

We thank the reviewer for this great idea. However, we did not manage to find a suitable solution for this task in the given timeframe of the revisions and believe pursuing this further would go beyond the scope of the manuscript. We did a few initial tests in order to identify covariates that would correlate with the occurrence probability of amplifications. Indeed, there are epigenomic properties of the tissue of origin-associated with copy number amplification breakpoints (see also Cramer, Serrano, Schaefer. *eLife*. 2016) but most of them (such as heterochromatin) are correlated with the mutation frequency too. As our work aims to demonstrate that the mutation frequency affects the selective pressure acting on amplification frequency, incorporating those would create a circularity that is not easy to control.

In addition, we did not find an easy way to define selectively neutral amplifications as would be required to normalize by the background amplification frequency directly (as done for the dN/dS).

Minor comments:

18.

p2: "common measures of selection – such as the ratio of non-synonymous to synonymous substitutions (dN/dS)".

Note that dN/dS is actually the ratio of non-synonymous to synonymous substitution rates.

We corrected this mistake on line 37.

19.

p3: "This phenomenon has been already observed in different species such as the asexual amoebas and bacteria".

It would be more correct to say "Similar phenomena", as those in amoebae and bacteria are not exactly the same process as somatic copy number amplification.

We incorporated the formulation of the reviewer on line 71.

20.

p3: "Moreover, whole-genome doubling – which preserves the ratio of mutant to wild-type alleles while doubling the number of wild-type ones – has been recently suggested to serve as a compensatory phenomenon for the accumulation of deleterious mutations in loss-of-heterozygosity regions".

Note that this argument applies to regions of "haploid LOH", and not so much to copy-neutral LOH (see López et al. 2020).

We corrected this mistake on line 81.

21.

p4: "considering only mutations within copy-neutral regions".

It would be useful to clarify in the Introduction that "copy-neutral" refers to regions present in two copies.

We now explain in the method section of the revised version of the manuscript the definition of copy neutrality. We refer the reviewer to lines 457-461.

22.

p6: " $j = 36$ Mbp turned out to be the optimal segment length for further analysis since it displayed the best associations in terms of Spearman's correlation and associated p-value".

It could be clarified that " j " refers to the total length between both ends of the amplified segment (if that is the case). The use of the label "Gene" in the x-axis of Fig 1e makes it unclear whether this instead represents distance in either direction from the gene locus.

We thank the reviewer for highlighting this issue as we agree that the visualization was not optimal before. We have addressed this by making two changes. Firstly, we have modified the x-axis label of Fig. 1e as suggested by reviewer #3 to "Segment length (j)". Secondly, we have increased the spacing between the "Gene" label and the "Bins (Mbp)" interval. This change has made it clear that "Gene" is as distinct (such are the "Arm" and "Chromosome" labels), and that the axis does not "represent distance in either direction from the gene locus".

23.

p7: "Our relationship analysis demonstrated that frequently mutated haploinsufficient genes tend to be amplified during tumor evolution, which is likely to create a buffer to compensate for the loss of the gene product upon mutation".

Of note, this finding is consistent with Martincorena et al.'s (2017) results for cancer genomes, where they observed no detectable negative selection in haploinsufficient genes (Fig 3H), and negative selection against mutations in essential genes located in haploid regions, but not in diploid regions (Fig 3G).

We added this specific reference to the new version of the manuscript on line 397-399.

24.

p8: "Hence, we removed from the analysis the mutations within OGs and TSGs to assess whether they may inflate the correlations, which, in the end, were neither significantly decreasing nor increasing across all cancer types (Fig. 2d and Fig. S3a ...)". This should be Fig. S3d, not S3a.

We corrected this mistake.

25.

p9: "Based on the analyses from Davoli et al. and Sack et al., we designed a modified version of the OG and GO scores ...". A brief mention of what these scores measure and what they mean should be included in the main text. The Methods contain the formulas for these scores, but do not explain what they mean (or even the meaning of the abbreviations "GO" and "OG"). Note also that "GO" is being used with two different meanings in the paper, which might cause confusion.

We refer the reviewer to question 6 of reviewer #2.

Moreover, we decided to be consistent and stick to Sack et al. nomenclature. We decide not to use the acronym GO for Gene Ontology, but only for referring to GO genes in the context of "STOP and GO" genes (defined in Sack et al., Cell, 2013).

26.

Table S3: Note that very small p-values are erroneously written with a positive exponent, e.g. "1.98E+10", rather than "1.98E-10".

We corrected the supplement and provided the new updated table.

Reviewer #5

Alfieri et al presented a study linking genomic amplifications with accumulation of deleterious mutations across the cancer genomes from TCGA. The authors found that regions of the genome that tend to accumulate more deleterious mutations often have higher frequencies of amplifications. They found haplo-insufficient and essential genes tend to have better protection from amplifications. They also explored this phenomenon in tissue specific manner. Subsequently, they dated the relative timing of amplifications and mutations. In the end, they performed functional enrichment analysis of the “essential” genes.

My biggest concern with this study is lacking of novelty. The relationship between genome instability/amplification and deleterious mutations have been known for many years, so, it is not a new hypothesis. The major novelty the authors developed is the correlation between local amplifications and mutations. Using this correlation metric, they explored potential connections to many different factors, each of which delivered findings with significant overlap with what we already know in the field and the statistical evidence are often not very strong and consistent across cancer types. In addition, the presentation is a collection of “positive results” from their explorations and the results are rather discrete. The link to existing literature is not as strong. I feel this might be a good paper for a professional journal.

We express our gratitude to the reviewer for providing us with valuable feedback. In particular, we feel that the revised version of our manuscript has substantially improved by addressing the two primary criticisms raised by the reviewer regarding the lack of consistency on cancer type level for some analyses and novelty of the research question. We have now highlighted previous research and debates that prompted our study of the buffering effect exerted by copy number amplifications, and we have also further explored the patterns that emerge on a cancer-type level, where the effects described in our study are weaker in some types of cancer. See our replies below. Overall, we believe that our analyses provide robust evidence for the existence of a buffering effect of copy number amplifications. Nonetheless, we have critically discussed both the strengths and limitations of our work in the Discussion to avoid giving the impression that our findings are beyond questioning.

Here are my major concerns:

1) The “gap in the field” in the introduction is not very clear. As I mentioned earlier, the hypothesis the authors articulated is known for many years.

While the hypothesis that amplifications could serve as buffers to protect cancer genomes from the detrimental impact of deleterious mutations has been proposed as an idea several times and has been specifically tested for WGDs in lung cancer (Lopez et al. Nat Gen. 2021), there is to our knowledge no systematic study to test for a general role of amplifications as buffers and the implications of this model (e.g., how this relates to tissue specificity of CNAs and if this would allow to detect cancer vulnerability genes). We expanded the introduction to more clearly highlight the novel aspects of our work and discuss the previous literature more rigorously. We refer the reviewer to the new version of the manuscript where we substantially expanded the Introduction section lines 65-71 and the Discussion section on lines 349-382.

2) When the authors performed segment specific correlation analysis, was multiple test correction performed?

We did not perform multiple testing correction when we compared cancer types or segment lengths as we merely wanted to identify cancer types / segments with maximal correlations and were dealing with a limited number of cancer types and segments. For the gene ontology analysis where we were specifically interested if and which terms are enriched in our set of protected and unprotected genes (and are performing a much larger number of tests), we performed multiple testing correction. We highlight this more clearly now in the manuscript on lines 589-590 and 598-599.

3) The haplo-insufficient genes, were discovered based on the same principle as the authors’ definition of compensatory effect. In some sense, the finding presented in Figure 2a is a bit circular.

We thank the reviewer for expressing their concern regarding the potential circularity of the findings presented in **Fig. 2a**. We would like to clarify that the pLI scores also incorporate mutation counts, but the calculation is fundamentally different. Therefore, we do not see how this could create a circularity or constitute a confounder. The calculation uses “the deviation of observed counts from the expected number of synonymous and missense mutations. Positive Z scores indicate increased constraints (intolerance to variation) and therefore that the gene had fewer variants than expected. Negative Z scores are given to genes that had more variants than expected

[...]”. This approach is based on the assumption that genes that are haploinsufficient or essential for cellular function will be more intolerant to variation.

Therefore, it is true that copy-number alterations were not taken into account for this calculation of pLI scores (we also refer the reviewer to <https://www.nature.com/articles/nature19057>).

Furthermore, we decided to apply the genome-wide haploinsufficiency score (GHIS). It is worth noting that (1) GHIS uses a different approach to predict the gene haploinsufficiency, and (2) there is not such a strong correlation between the pLI and the GHIS scores (Spearman’s correlation of 0.356). We kindly refer the reviewer to the revised version of the manuscript in the Results section on lines 174-176 and in the Method section on lines 492-501 and main **Fig. 2**.

4) The results surrounding Figure 2b (conservation score such as Polyphen and CADD) and Figure 2c (aggregating regions) are not consistent across cancer types. The discrepancies across cancer types significantly challenge the authors’ hypothesis and conclusion.

We added a disclaimer in the Discussion specifically stating that there are some exceptions (lines 401-406):

“It is worth noting that despite the clear tendency observed for most of the correlations across cancer types, in some specific situations, we do not observe the general trend. We would like to point out, however, that many of the gene and mutation properties (such as mutation impact or aggregation propensity) are predictions based on various features, partly with limitations in accuracy or being highly tissue specific.”

5) The section on tissue specific amplification pattern is very unclear and very hard to follow.

We apologize to the reviewer if the initial version of the tissue-specificity section was unclear and difficult to follow. Our analysis aimed to examine tissue-specific aneuploidy and amplification patterns, which are common in cancer cells. Based on previous research, we know that these patterns are often associated with the presence of oncogenes and genes that induce proliferation in specific tissue types (referred to as GO genes from Sack et al.). Here, we aimed to investigate whether the μ score might also contribute to the tissue-specificity of amplifications.

We revised this section, and we hope that it is now clearer and easier to follow (see lines 230-235, and some other minor changes).

6) Timing of the mutational events (amplifications and mutations), the hypothesis isn’t as clear. As mutations are continuously accumulating (not a single event), the hypothesis the authors articulated is not as clear. In addition, the timing results for different karyotypes are very different in Figure 4. I am not sure the results are robust and consistent.

We hope that the new version of the manuscript may better address how we performed the timing of mutational events (Results section on lines 293-309, Method section on lines 545-552 and new **Fig. 4**) and why it is necessary for our investigation. Moreover, we have revised some ambiguous sentences that could lead to misinterpretations of the temporal order of events (we also refer the reviewer to comment #2 from reviewer #4).

7) It is hard to sense the novelty from the functional enrichment of the protected gene list identified in the last section.

We would like to clarify that the functional enrichment test of protected genes is not an exploratory analysis, but rather a confirmatory test aimed to validate our model. We argue that cancer-essential functions, in particular, are buffered by genomic amplifications and therefore protected from deleterious mutations. However, we envision using in the future an approach similar to the one described here to identify protected genes that might be cancer vulnerabilities. Since this would require experimental validation of the identified targets, we have decided to refrain from including such an analysis.

REVIEWERS' COMMENTS

Reviewer #2 (Remarks to the Author):

The authors provided an extensive rebuttal and added several new analyses. I have no further comments and wish to congratulate the authors with their work.

Reviewer #3 (Remarks to the Author):

The authors have addressed my previous concerns and I have no more comments for them.

Reviewer #4 (Remarks to the Author):

I congratulate Alfieri and colleagues on their effort to address the concerns raised by myself and the other reviewers. I believe their revision has substantially improved the results and framing of this study. I consider my previous concerns, especially regarding the ambiguity of the paper's theoretical framework and methods, satisfactorily addressed. I have a small number of further minor comments, however, which are provided below in order of importance. These should be easy to address, and I would regard the manuscript as apt for publication provided that they are satisfactorily addressed.

1. Lines 176, 186, 195, 206, 220, 226, 298, 331, 334, 337, 591:

On re-reading the manuscript, I am concerned by the way in which one-sided tests are used throughout without a clear justification for this choice. As one-sided tests inherently give lower p-values than the corresponding two-sided tests, they should always be used carefully, and their use should be well justified. That is, if the authors have a valid reason for the unusual choice of using only one-sided alternative hypotheses, they should disclaim and justify this decision in the manuscript. At any rate, it seems that it would be safer to use two-sided tests, as one-sided tests can inflate statistical significance (increasing false positive rates) and are sometimes viewed as a tool for 'p-hacking'.

Moreover, if the authors decide to retain one-sided testing, then the text should be revised carefully to avoid invalid claims on the basis of such tests. For instance, the manuscript repeatedly refers to "testing for differences", despite the fact that the employed tests are often testing only for an increase in correlation, rather than general differences. Therefore, not only do these tests give smaller p-values

than would be obtained with two-sided tests, but their hypotheses are also narrower than is often implied in the text.

To give a specific example, line 225 says that correlations "were neither significantly decreasing nor increasing across all cancer types", but this claim cannot be justified using one-sided tests, as the latter only test for differences in one direction. Even though a look at the data suggests that a two-sided test would lead to the same conclusion, the point is that 'bidirectional' statements of this kind are not statistically valid when using one-sided tests.

Therefore, I would advise the authors either to use two-sided tests (which are more standard and convincing) or to revise the text very carefully to acknowledge and reflect the fact that they are often testing for differences in a specific direction (i.e. using one-sided alternative hypotheses).

2. p4, l94: "Eq. 1 in Fig. S1 and Methods"

There is no Eq. 1 (only 1a and 1b). This presumably refers to Eq. 2; please correct.

3. p12, l291: "In contrast, segment's allele-specific copy numbers where the minor allele provides at least two unmutated copies of the allele do not show a significant association between late mutations and coding mutations (i.e., x:2 and x:3 cases)."

Except for 3:3 (according to Fig. 4a).

4. p22, l526-536:

In the previous round of review, I noted that the OG and GO scores are defined and employed in the paper without a proper description of what they are and what they measure. This does not seem to have been addressed by the authors. A very brief description of these scores in this section of the Methods would be helpful. Moreover, the meaning of the terms in Eqs. 3 and 4 ($OG_{i,j}$; $GO_{i,j}$; $genes_{i,j}$) should be explained.

5. p1, l11: "While the majority of CODING mutations are deleterious"

Please correct.

6. p6, l125: There is a missing "." before "Lung Squamous cell Carcinoma".

7. As a suggestion: When writing "allele-specific copy number", note that the abbreviation "CN" is available for copy number. Besides, an even shorter abbreviation could be introduced the first time this expression is used, for instance: "a segment's allele-specific copy number (hereafter, segment CN)".

8. Note that refs. 5 and 28 are the same.

Reviewer #5 (Remarks to the Author):

Thanks for addressing my questions.

Reviewer #2

The authors provided an extensive rebuttal and added several new analyses. I have no further comments and wish to congratulate the authors with their work.

We thank the reviewer, and we are grateful for the improvements they suggested.

Reviewer #3

The authors have addressed my previous concerns and I have no more comments for them.

We thank the reviewer, and we are grateful for the improvements they suggested.

Reviewer #4

I congratulate Alfieri and colleagues on their effort to address the concerns raised by myself and the other reviewers. I believe their revision has substantially improved the results and framing of this study. I consider my previous concerns, especially regarding the ambiguity of the paper's theoretical framework and methods, satisfactorily addressed. I have a small number of further minor comments, however, which are provided below in order of importance. These should be easy to address, and I would regard the manuscript as apt for publication provided that they are satisfactorily addressed.

We thank the reviewer, and we are grateful for the improvements they suggested.

1. Lines 176, 186, 195, 206, 220, 226, 298, 331, 334, 337, 591:

On re-reading the manuscript, I am concerned by the way in which one-sided tests are used throughout without a clear justification for this choice. As one-sided tests inherently give lower p-values than the corresponding two-sided tests, they should always be used carefully, and their use should be well justified. That is, if the authors have a valid reason for the unusual choice of using only one-sided alternative hypotheses, they should disclaim and justify this decision in the manuscript. At any rate, it seems that it would be safer to use two-sided tests, as one-sided tests can inflate statistical significance (increasing false positive rates) and are sometimes viewed as a tool for 'p-hacking'.

Moreover, if the authors decide to retain one-sided testing, then the text should be revised carefully to avoid invalid claims on the basis of such tests. For instance, the manuscript repeatedly refers to "testing for differences", despite the fact that the employed tests are often testing only for an increase in correlation, rather than general differences. Therefore, not only do these tests give smaller p-values than would be obtained with two-sided tests, but their hypotheses are also narrower than is often implied in the text.

To give a specific example, line 225 says that correlations "were neither significantly decreasing nor increasing across all cancer types", but this claim cannot be justified using one-sided tests, as the latter only test for differences in one direction. Even though a look at the data suggests that a two-sided test would lead to the same conclusion, the point is that 'bidirectional' statements of this kind are not statistically valid when using one-sided tests.

Therefore, I would advise the authors either to use two-sided tests (which are more standard and convincing) or to revise the text very carefully to acknowledge and reflect the fact that they are often testing for differences in a specific direction (i.e. using one-sided alternative hypotheses).

We agree with the reviewer's concern, and we replaced all one-sided tests in the revised version of the manuscript with two-sided ones. This did not change our conclusions as all previously significant tests remained significant. This affects lines 117, 186, 196, 207, 221 and 228 and Figure 2.

2. p4, l94: "Eq. 1 in Fig. S1 and Methods"

There is no Eq. 1 (only 1a and 1b). This presumably refers to Eq. 2; please correct.

We corrected this mistake.

3. p12, l291: "In contrast, segment's allele-specific copy numbers where the minor allele provides at least two unmutated copies of the allele do not show a significant association between late mutations and coding mutations (i.e., x:2 and x:3 cases)."

Except for 3:3 (according to Fig. 4a).

We added this on line 294.

4. p22, l526-536:

In the previous round of review, I noted that the OG and GO scores are defined and employed in the paper without a proper description of what they are and what they measure. This does not seem to have been addressed by the authors. A very brief description of these scores in this section of the Methods would be helpful. Moreover, the meaning of the terms in Eqs. 3 and 4 ($OG_{i,j}$; $GO_{i,j}$; $genes_{i,j}$) should be explained.

We added an explanatory sentence on line 533.

5. p1, l11: "While the majority of CODING mutations are deleterious"
Please correct.

We corrected this.

6. p6, l125: There is a missing ":" before "Lung Squamous cell Carcinoma".

We corrected this mistake.

7. As a suggestion: When writing "allele-specific copy number", note that the abbreviation "CN" is available for copy number. Besides, an even shorter abbreviation could be introduced the first time this expression is used, for instance: "a segment's allele-specific copy number (hereafter, segment CN)".

We used the available abbreviation CN for copy number as the reviewer suggested. For a matter of clarity, we have decided to not further abbreviate "segment allele-specific copy number".

8. Note that refs. 5 and 28 are the same.

We corrected this mistake.

Reviewer #5

Thanks for addressing my questions.

We thank the reviewer, and we are grateful for the improvements they suggested.